# Towards a Multimodal Foundation Model for Time Series Analysis

## Abstract

Time series analysis supports a wide range of real-world applications. While existing time series foundation models primarily rely on large-scale unimodal pretraining, they lack complementary modalities to enhance time series understanding. Building multimodal foundation models is a natural next step, but it introduces key challenges: 1) lack of a unified multimodal pretraining paradigm and large-scale multimodal corpora for time series analysis; 2) how to effectively integrate heterogeneous modalities and enhance model generalization across both modalities and domains. To address these challenges, we take an early step toward multimodal foundation models for time series analysis. We first propose a multimodal pretraining paradigm that leverages time series together with their derived image and text, enhancing time series analysis from a multi-view perspective. Building upon this paradigm , we construct MM-TS, a large-scale multimodal dataset spanning time series, text, and image across six domains, with more than one billion time points. Then we propose HORAI, a frequency-enhanced multimodal foundation model. HORAI integrates two core components: a Frequency-guided Cross-Modality Encoder, which leverages the correspondence between modality-specific information and different frequency components of time series to effectively fuse multiple modalities, and a Time-Frequency Decoder, which incorporates frequency information into a MoE router to improve pattern discrimination and generalization. After pretraining on MM-TS, HORAI achieves state-of-the-art zero-shot performance on time series forecasting and anomaly detection tasks, demonstrating strong task versatility and generalization.

## 1 Introduction

Time series analysis is widely applied across diverse domains, including energy management, medical monitoring, and financial forecasting. Existing time series analysis approaches, ranging from time-series-specific models (Zeng et al., 2023; Nie et al., 2023; Liu et al., 2024b; Chen et al., 2024b) to recent time series foundation models (Liu et al., 2024c; Woo et al., 2024; Gao et al., 2024; Shi et al., 2025; Wang et al., 2025b), primarily rely on time series numerical modality to capture temporal patterns and uncover underlying regularities. While these methods have achieved competitive performance, this single-modality paradigm remains limited in its ability to capture the complex and multifaceted nature of real-world temporal dynamics (Xu et al., 2024a).

At the same time, foundation models in NLP and multimodal learning (Brown et al., 2020; Bai et al., 2023; Wu et al., 2024; Chen et al., 2024c) have shown that large-scale pretraining on massive datasets with complementary modalities can enhance generalization and adaptability across tasks. Inspired by these, we propose developing multimodal foundation models for time series analysis. By incorporating additional modalities for pretraining, such as texts and images, the model leverages textual semantics and visual, spatial information to better capture complex temporal dynamics and strengthen time series understanding.

However, the development of multimodal foundation models faces several significant challenges. First, lack of a unified multimodal pretraining paradigm and large-scale multimodal corpora for time series analysis. Multimodal pre-training for time series remains in a nascent stage. Existing methods are either restricted to end-to-end training on small-scale multimodal datasets or confined to large-scale unimodal pre-training due to the scarcity of aligned modalities. Therefore, establish-

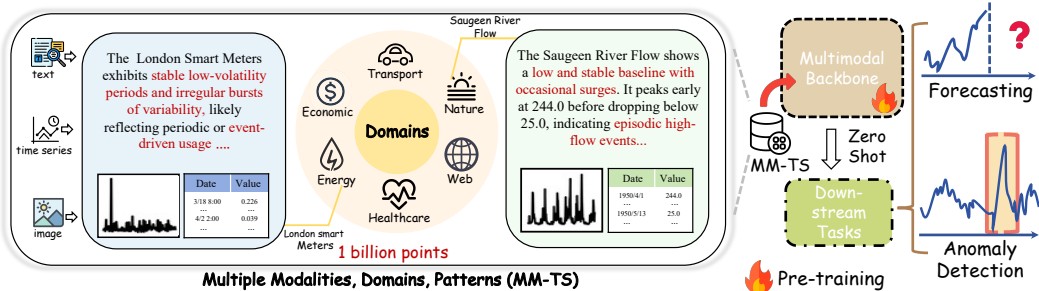

Figure 1: Left: The large-scale multimodal time series dataset MM-TS is characterized by its coverage of various modalities, heterogeneous domains, and diverse temporal patterns. Right: The multimodal foundation model (HORAI) is pre-trained on the MM-TS dataset and evaluated on downstream scenarios and tasks.

ing an effective pre-training paradigm alongside large-scale, well-aligned datasets is indispensable for advancing multimodal pre-training. Second, the architectural design for integrating different modalities in time series analysis remains underexplored. Each modality exhibits unique characteristics: text exhibits rich semantic information and provides a holistic, global description of events, whereas image captures localized details and spatial structures (Zhong et al., 2025). Directly fusing time series with textual or visual modalities (Phuong & Lampert, 2019; Kim & Rush, 2016) may result in suboptimal alignment and ineffective representation learning. Therefore, it is critical to design fusion mechanisms that explicitly leverage the unique characteristics of each modality. Third, time series data from different domains exhibit diverse patterns, and the incorporation of multiple modalities further amplifies this diversity. Effectively modeling the heterogeneous patterns across modalities and domains, while enhancing the generalization ability of pretrained models, remains a challenge. Consequently, advancing multi-modal foundation models for time series analysis requires further research and exploration.

In this paper, we take an early step toward developing multimodal foundation models for time series analysis. *On the pretraining paradigm and dataset side*, we propose a novel paradigm that utilizes three time series, images, and text as three modalities by an endogenous construction strategy. This approach synthesizes large-scale aligned multimodal data to enhance time series analysis from a multi-view perspective, leveraging endogenous pre-training to adapt to exogenous modalities, thereby enabling good zero-shot generalization in downstream scenarios. Based on this paradigm, we construct the first large-scale multimodal time series dataset (MM-TS). As illustrated in Figure 1, different from existing time series datasets, MM-TS integrates three modalities, including time series, text, and images, spanning six diverse domains and a wide range of temporal patterns, with up to one billion time series points. The three modalities exhibit strong correlations and complementary characteristics, making MM-TS well-suited for multimodal pretraining to learn generalized representations. This dataset provides a solid foundation for studying multimodal models.

*On the modeling side*, we propose **HORAI**, a frequency-enhanced multimodal time series foundation model built on an autoregressive architecture, which consists of two core components: Frequency-guided Cross-Modality Encoder and Time-Frequency Decoder. *In the Frequency-guided Cross-Modality Encoder*, we leverage the correspondence between modality-specific information and different frequency components of time series to align multiple modalities and enhance time series understanding. Specifically, time series are decomposed into multiple frequency bands, where low-frequency components capture long-term dynamics and align with the global semantics embedded in text, while mid- and high-frequency components encode rapid variations that tend to correspond to the localized patterns present in visual inputs. Given the large number of tokens in text and image modalities, we further incorporate the flow-attention alignment mechanism to facilitate efficient cross-modal alignment while preserving the fidelity of features. *In the Time-Frequency Decoder*, we design a Time-Frequency MoE-FFN to learn generalized multimodal representations from multi-domain data. We introduce a time-frequency router that dynamically assigns each token to the suitable expert based on both its temporal and frequency features. By incorporating frequency-domain features, the router gains additional cues to better distinguish similar patterns and group them coherently, which enhances feature consistency and improves generalization across domains and modalities. As shown in Figure 1, HORAI is pre-trained on the MM-TS dataset and evaluated on

various downstream scenarios and tasks: forecasting and anomaly detection, demonstrating strong generalization capabilities. Specially, our contributions can be summarized as follows:

- We propose a multimodal pretraining paradigm that leverages time series together with their derived image and text, enhancing time series analysis from a multi-view perspective. Building upon this paradigm, we delve into the multimodal time series foundation model development by constructing a large-scale multimodal time series pretraining dataset (MM-TS), which covers six diverse domains and three modalities, with up to 1 billion time points.

- We propose HORAI, a frequency-enhanced multimodal foundation model for Time series analysis, which incorporates two core components, the frequency-guided cross-modality encoder and the time-frequency decoder, designed to effectively fuse multimodal features and enhance model generalization across modalities and domains.

- After pre-training on large-scale multimodal time series data, HORAI achieves state-of-the-art performance in time series forecasting and anomaly detection across zero-shot inference and few-shot learning situations, which demonstrates strong task versatility and generalization ability.

## 2 RELATED WORK

### 2.1 TIME SERIES ANALYSIS

Time series analysis spans a wide range of tasks, including forecasting and anomaly detection (Qiu et al., 2024; Faloutsos et al., 2018; Darban et al., 2025; Paparrizos et al., 2022b). Existing approaches can be broadly divided into unimodal and multimodal methods. Unimodal methods focus on time series data and employ diverse architectures to model temporal dynamics and channel correlations. These include MLP-based models (Zeng et al., 2023; Xu et al., 2024b; Zhong et al., 2024), RNN-based models (Flunkert et al., 2017; Cirstea et al., 2019), CNN-based models (Wu et al., 2023; Luo & Wang, 2024), GNN-based models (Zhao et al., 2023; Wu et al., 2021), as well as Transformer-based architectures for capturing long-range dependencies (Zhang & Yan, 2023; Nie et al., 2023; Chen et al., 2024b; Yang et al., 2023). In contrast, multimodal methods integrate additional modalities or external knowledge to enhance time series analysis. One line of work introduces endogenous prompts, such as statistical information, channel semantics, or task-related descriptions, to enrich temporal representations (Jin et al., 2024; Chen et al., 2025; Pan et al., 2024; Zhong et al., 2025). Another line of work leverages exogenous textual or visual modalities to provide additional contextual knowledge (Li et al., 2025; Jia et al., 2024; Wang et al., 2025a; Liu et al., 2024a). Although these methods achieve competitive performance, most require retraining and extensive parameter tuning for each dataset, lacking zero-shot inference capabilities. While ChatTime (Wang et al., 2025a) enables direct zero-shot inference, it suffers from precision loss due to data discretization and lacks rich multimodal characterizations.

### 2.2 TIME SERIES FOUNDATION MODELS

Foundation models pre-trained on large-scale data have achieved notable success in language (Brown et al., 2020; Touvron et al., 2023) and vision (Liu et al., 2021; Dosovitskiy et al., 2021) domains. Recently, time series foundation models (TSFMs) have attracted increasing attention (Liu et al., 2024c; Ansari et al., 2024; Woo et al., 2024; Das et al., 2024; Goswami et al., 2024; Ekambaram et al., 2024; Chen et al., 2024a; Shi et al., 2025; Liu et al., 2025). By pre-training on large-scale and diverse time series datasets, these models exhibit strong adaptability to new tasks, enabling both efficient fine-tuning and zero-shot transfer across domains. For instance, Timer (Liu et al., 2024c) employs a decoder-only architecture with autoregressive pre-training to capture temporal dependencies, while MOIRAI (Woo et al., 2024) introduces multi-scale patch projections to model diverse patterns and an any-variate attention mechanism that allows flexible handling of time series with arbitrary dimensionality. ROSE (Wang et al., 2025b) combines frequency decomposition with time-series registers to jointly learn both domain-invariant and domain-specific representations, facilitating knowledge transfer to downstream tasks. Sundial (Liu et al., 2025) proposes a TimeFlow Loss that predicts the distribution of the next patch, enabling Transformer training without discrete tokenization while supporting probabilistic forecasting.

Existing TSFMs are all pre-trained solely on unimodal time series data, which provides some generalization ability but cannot leverage complementary modalities to model more complex temporal dynamics. In contrast, HORAI effectively leverages multiple modalities through a frequency-enhanced cross-modality encoder and introduces a Time-Frequency Decoder to further strengthen cross-modality and cross-domain generalization during pre-training.

## 3 METHODOLOGY

### 3.1 LARGE-SCALE MULTIMODAL TIME SERIES DATASET

Large-scale datasets are the cornerstone of pre-training foundation models, enabling them to acquire transferable knowledge and improve generalization across diverse downstream scenarios. However, existing large-scale time series corpora are mostly confined to unimodal time series data, which limits the potential of multimodal learning. To address this problem, we conduct MM-TS, a large-scale multimodal time series dataset for pre-training. As shown in Figure 1, MM-TS integrates three modalities: time series, text, and image, covering six diverse domains, including Energy, Healthcare, Web, Nature, Transport, and Economics. In total, MM-TS contains over 1 billion time points, setting a new scale for multimodal time series research.

For the time series modality, MM-TS spans multiple temporal granularities such as seconds, minutes, hours, and months, and captures diverse characteristics including periodicity, trends, and non-stationarity (see Appendix A.1 for details). For the textual modality, due to the scarcity of natural paired descriptions, we design prompts and leverage large language models to generate semantic descriptions. These descriptions capture temporal dynamics, for example, "stable low-volatility periods with irregular bursts of variability," and also provide causal reasoning, such as attributing sudden growth to event-driven factors. For the visual modality, we construct line-plot images directly from time series, offering an intuitive view of temporal fluctuations and structural patterns.

By unifying multimodal data across domains, MM-TS provides a high-quality, large-scale resource for scalable multimodal pre-training, paving the way toward foundation models for time series analysis with generalization capabilities.

### 3.2 HORAI

To better leverage cross-modal and cross-domain features for enhanced time series understanding, we propose HORAI, a frequency-enhanced multimodal foundation model for time series analysis. HORAI consists of two core components: the Frequency-guided Cross-Modality Encoder and the Time-Frequency Decoder. As illustrated in Figure 2, in the cross-modality encoder, the input time series is first decomposed into low-frequency and mid-to-high-frequency components, which are aligned with textual and visual features, respectively. Then, an adaptive modality fusion module subsequently combines these aligned representations to produce unified multimodal representations. In the Time-Frequency Decoder, the multimodal representations are first passed into a Time-Frequency MoE-FFN, which is designed to capture diverse patterns across multiple domains. To guide the routing of tokens to appropriate experts, both temporal-domain and frequency-domain features are incorporated. The inclusion of frequency information provides additional cues that help distinguish similar patterns and group them coherently, enhancing the model's cross-modality and cross-domain generalization. Finally, the learned representations are projected through a token projection layer for autoregressive pre-training.

#### 3.2.1 FREQUENCY-ENHANCED CROSS-MODAL ENCODER

**Multimodal Embedding.** For notational simplicity, we describe the method using a univariate time series, which can be easily extended to the multivariate case by treating each channel independently. Given an input time series $\mathbf{X}_{\mathrm{ts}} \in \mathbb{R}^T$, where $T$ denotes the sequence length, we first apply instance normalization (Kim et al., 2021) to mitigate distribution shift, resulting in $\mathbf{X}_{\mathrm{norm}} \in \mathbb{R}^T$.

Since different frequency components capture different aspects of temporal dynamics, with low-frequency components reflecting global trends and mid-to-high-frequency components capturing local variations, we transform the normalized sequence into the frequency domain by the Fast Fourier Transform (FFT), obtaining $\mathbf{X}_{\mathrm{freq}} \in \mathbb{R}^{L/2+1}$. To separate different frequency bands, we set a ratio

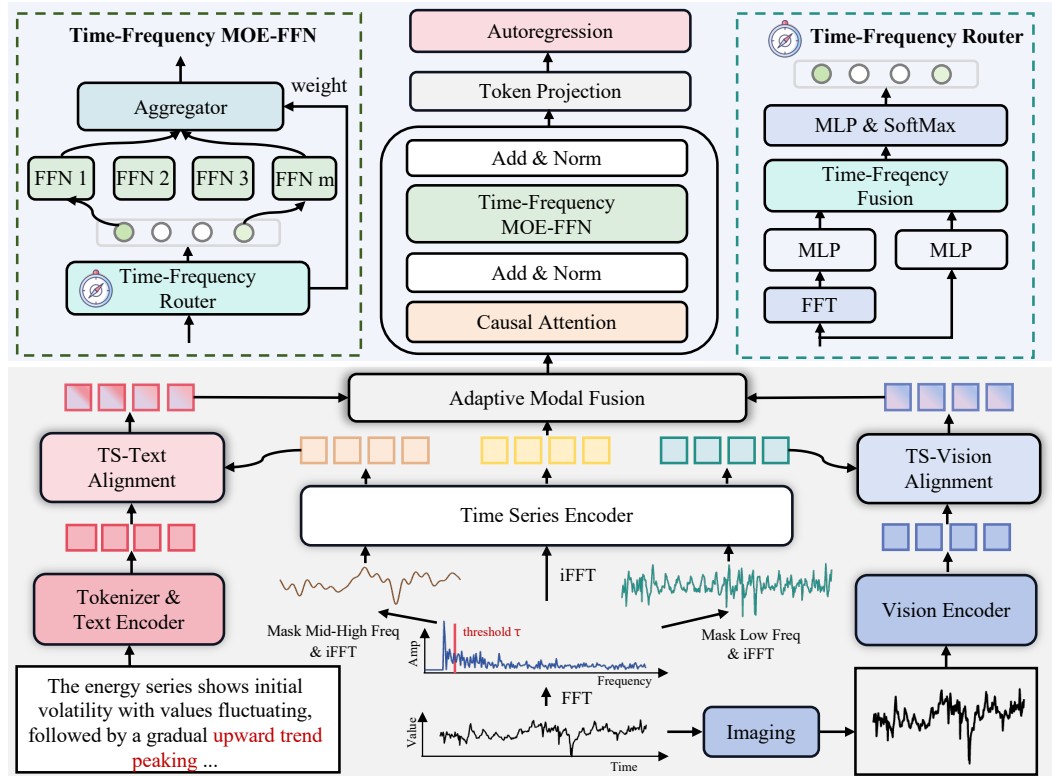

Figure 2: The framework of the proposed HORAI consists of a Frequency-Enhanced Cross-Modal Encoder (gray region) and a Time-Frequency Decoder (blue region).

parameter $\alpha$ to define a cutoff threshold $\tau = \alpha \cdot (L/2+1)$. Based on this threshold, we construct two binary masks: $\mathbf{M}_{\text{low}} \in \{0,1\}^{L/2+1}$ for low-frequency components and $\mathbf{M}_{\text{mh}} \in \{0,1\}^{L/2+1}$ for mid-to-high-frequency components. Applying these masks to $\mathbf{X}_{\text{freq}}$ by element-wise multiplication yields two masked spectra, which are then transformed back into the time domain using the inverse FFT (iFFT). This process produces the low-frequency sequence $\mathbf{X}_{\text{low}} \in \mathbb{R}^L$ and the mid-to-high-frequency sequence $\mathbf{X}_{\text{mh}} \in \mathbb{R}^L$.

$$\mathbf{X}_{\text{low}} = \text{iFFT}(\mathbf{X}_{\text{norm}} \odot \mathbf{M}_{\text{mh}}), \quad \mathbf{X}_{\text{mh}} = \text{iFFT}(\mathbf{X}_{\text{norm}} \odot \mathbf{M}_{\text{low}}). \tag{1}$$

Subsequently, we employ a patching strategy to divide $\mathbf{X}_{\text{low}}$, $\mathbf{X}_{\text{mh}}$, and $\mathbf{X}_{\text{norm}}$ into $N_{ts}$ patches with patch size $S$. These patches are projected and fed into the time-series encoder (Nie et al., 2023), producing corresponding time series representations: $\mathbf{E}_{\text{low}}$, $\mathbf{E}_{\text{mh}}$, and $\mathbf{E}_{\text{ts}} \in \mathbb{R}^{N_{ts} \times D_{ts}}$.

For the textual input $\mathbf{X}_{\text{text}} \in \mathbb{R}^{L_{\text{text}}}$, we employ a text tokenizer followed by a pre-trained text encoder to extract semantic features, yielding $\mathbf{E}_{\text{text}} \in \mathbb{R}^{L_{\text{text}} \times D_{\text{text}}}$. For the visual input $\mathbf{X}_{\text{img}} \in \mathbb{R}^{C \times H \times W}$, we apply a patching strategy and a pre-trained vision encoder to obtain image representations $\mathbf{E}_{\text{img}} \in \mathbb{R}^{N_{\text{img}} \times D_{\text{img}}}$.

**Frequency-enhanced Cross-Modality Alignment.** Time series often exhibit rich frequency-dependent patterns, where low-frequency components capture global trends and mid-to-high-frequency components reflect local variations. Meanwhile, different modalities contribute differently to these patterns: textual information tends to describe global trends, aligning with low-frequency time series components, whereas visual information focuses more on short-term variation, corresponding to mid-to-high-frequency components (Zhong et al., 2025). Motivated by this, we propose a frequency-enhanced cross-modal fusion that explicitly leverages the characteristic correspondence between modalities and frequency components. Additionally, given the large number of tokens in text and image modalities, we integrate a Flow-Attention-based alignment mechanism to efficiently model cross-modal interactions while preserving the fidelity of features.

In the TS-Text Alignment module, the low-frequency time series embeddings and textual embeddings are first projected by MLPs into a shared representation space $D_{model}$. Cross-modal fusion is then performed efficiently using the Flow-Attention mechanism. The core idea is to treat attention as a flow of information and leverage the flow conservation principle to optimize the transmission and aggregation of features across modalities. Specifically, the low-frequency time series embeddings $\mathbf{E}_{\text{low}}$ are mapped to serve as the Query $\mathbf{Q}$, while the textual embeddings $\mathbf{E}'_{\text{text}}$ are mapped to serve as the Key $\mathbf{K}$ and Value $\mathbf{V}$. The information flow between tokens is computed as:

$$\mathbf{I}_i = \phi(\mathbf{Q}_i) \sum_{j=1}^{N_{\text{text}}} \phi(\mathbf{K}_j)^T, \quad \mathbf{O}_j = \phi(\mathbf{K}_j) \sum_{i=1}^{N_{\text{ts}}} \phi(\mathbf{Q}_i)^T, \quad \hat{\mathbf{O}} = \phi(\mathbf{K}) \sum_{i=1}^{N_{\text{ts}}} \frac{\phi(\mathbf{Q}_i)^T}{\mathbf{I}_i}, \tag{2}$$

$$\mathbf{E}'_{\text{text}} = \frac{\phi(\mathbf{Q})}{\mathbf{I}} (\phi(\mathbf{K})^T (\text{Softmax}(\hat{\mathbf{O}}) \odot \mathbf{V})),$$

$\phi(\cdot)$ denotes the non-linear projection to the flow space, $\mathbf{I}_i$ and $\mathbf{O}_j$ represent the total outgoing and incoming flows for each token. The output $\mathbf{E}'_{\text{text}} \in \mathbb{R}^{N_{ts} \times D_{model}}$ is a flow-attention enhanced textual embedding, which has been adaptively aligned with the low-frequency time-series features.

Similar to the low-frequency time-series and text fusion, the TS-Vision Alignment module also leverages the Flow-Attention mechanism to integrate mid-to-high-frequency time-series embeddings $\mathbf{E}_{\text{mh}}$ with image embeddings $\mathbf{E}_{\text{img}}$, yielding aligned image representations $\mathbf{E}'_{\text{img}} \in \mathbb{R}^{N_{ts} \times D_{model}}$ for subsequent multimodal fusion.

**Adaptive Modal Fusion.** Considering that the contributions of image and text representations vary across different time series patterns, we adaptively fuse the aligned image and text embeddings. The aligned image embeddings $\mathbf{E}'_{\text{img}}$ and text embeddings $\mathbf{E}'_{\text{text}}$ are concatenated along the feature dimension and then passed through a linear projection followed by a sigmoid function $\sigma$ to perform gated weighting $\mathbf{G}$, producing the multimodal representation $\mathbf{E}_{\text{mm}}$. This representation is subsequently added to the time series embeddings $\mathbf{E}_{\text{ts}}$ to obtain the fused representation $\mathbf{E}_{\text{fused}} \in \mathbb{R}^{N_{ts} \times D_{model}}$. The specific process is as follows:

$$\mathbf{G} = \sigma(W_g[\mathbf{E}'_{\text{image}}, \mathbf{E}'_{\text{text}}] + b_g), \quad \mathbf{E}_{\text{fused}} = \mathbf{G} \odot \mathbf{E}'_{\text{image}} + (1 - \mathbf{G}) \odot \mathbf{E}'_{\text{text}} + \mathbf{E}_{\text{ts}}. \tag{3}$$

### 3.2.2 Time-Frequency Decoder

Large-scale time series data inevitably involves diverse domains, which gives rise to a wide variety of temporal patterns (Wang et al., 2025b; Woo et al., 2024). The incorporation of textual and visual modalities further amplifies the diversity. To address this challenge, we propose a Time-Frequency Decoder designed to capture and adapt to different patterns, enhancing the generalization ability of pre-trained models. As illustrated in Figure 2, the Time-Frequency Decoder consists of key components including Causal Attention, Normalization layers, and a Time-Frequency MoE-FFN.

**Time-Frequency MoE-FFN.** Different expert networks can capture distinct patterns from large-scale data, so effectively routing multimodal features to the appropriate experts is crucial. However, relying only on temporal-domain features may lead to entangled representations across different patterns, which makes pattern discrimination less straightforward. By incorporating frequency-domain features, similar patterns can be represented more compactly, offering additional cues for more accurate expert routing. Motivated by this, we propose the Time-Frequency Router, which integrates both temporal and frequency information to enhance the routing process.

Based on the fused multi-modal representation $\mathbf{E}_{\text{fused}}$, we obtain representation $\mathbf{H}$ through causal attention followed by normalization. In the router, each token of $\mathbf{H}$ is projected in parallel across both temporal and frequency domains: (i) an MLP produces temporal representations $\mathbf{H}_{\text{temp}}$, while (ii) an FFT followed by an MLP yields frequency representations $\mathbf{H}_{\text{freq}}$. These dual-domain signals are adaptively integrated via a learnable gating function $G_{\text{router}}$, resulting in router representation $\mathbf{H}_r \in \mathbb{R}^{N_{ts} \times D_{model}}$:

$$\mathbf{H}_r^i = \mathbf{G}_{\text{router}} \odot \text{MLP}(\mathbf{H}^i) + (1 - \mathbf{G}_{\text{router}}) \odot \text{MLP}(\text{FFT}(\mathbf{H}^i)), \quad i = 1, \cdots, N_{ts}. \tag{4}$$

Given $\mathbf{H}_r$, the router applies an MLP-based routing function to compute routing weights $\mathbf{W} \in \mathbb{R}^M$, which determine expert assignment. Following a Top-K strategy, the router selects the $K$ experts

with the highest weights, denoting the set of their indexes as $\mathcal{K}$. Then their outputs are aggregated through weight-normalized fusion, producing the representation $\mathbf{H}_{\mathrm{moe}} \in \mathbb{R}^{N_{ts} \times D_{model}}$:

$$\mathbf{H}_{\mathrm{moe}}^{i} = \sum_{j \in \mathcal{K}} \frac{\exp(\mathbf{W}_j)}{\sum_{m \in \mathcal{K}} \exp(\mathbf{W}_m)} \mathrm{FFN}_j(\mathbf{H}^i), \quad i = 1, \cdots, N_{ts}. \tag{5}$$

**Autoregressive Training.** Given the strong performance of the autoregressive paradigm in both NLP Bai et al. (2023); Brown et al. (2020) and time series domains (Liu et al., 2025; 2024c), we adopt a GPT-style training objective to predict the next token. This autoregressive formulation not only supports variable input and output lengths flexibly during inference but also excels at iterative, multi-step generation. Specifically, each input token $\mathbf{X}_i \in \mathbb{R}^S$ is processed through the encoder, decoder, and token projection layer to generate the prediction of the subsequent token $\hat{\mathbf{X}}_{i+1} \in \mathbb{R}^S$. The overall optimization objective is defined as:

$$\mathcal{L}_{train} = \frac{1}{N_{ts}S} \sum \|\hat{\mathbf{X}}_i - \mathbf{X}_i\|^2, \quad i = 1, \ldots, N_{ts}. \tag{6}$$

## 4 EXPERIMENTS

### 4.1 EXPERIMENTAL SETUP

**Datasets.** We perform pre-training of HORAI on our proposed MM-TS dataset and *ensure no overlap between the pre-training MM-TS dataset and the downstream evaluation datasets*. To assess HORAI's capability for time series analysis, we use the widely used evaluation datasets (Liu et al., 2024a) for forecasting and anomaly detection tasks, including Climate, Energy, Environment, Social Good, Traffic, EWJ, KR, MDT, and Weather. Specific dataset information is in Appendix A.

**Baselines.** We select both time series foundation models and time-series-specific models of each task as baselines. *For the forecasting task*, we select five SOTA foundation models: ChatTime (Wang et al., 2025a), VisionTS (Chen et al., 2024a), ROSE (Wang et al., 2025b), Timer (Liu et al., 2024c), MOIRAI (Woo et al., 2024), and four *multimodal time-series-specific models*: GPT4MTS (Jia et al., 2024), TATS (Li et al., 2025), GPT4TS (Zhou et al., 2023), TimeVLM (Zhong et al., 2025). *For the anomaly detection task*, we select three unimodal foundation models: DADA (Shentu et al., 2025), Timer, UniTS (Gao et al., 2024), and nine time-series-specific models: GPT4TS, LLMMixer (Kowsher et al., 2024), TimesNet (Wu et al., 2023), DCdetector (Yang et al., 2023), Anomlay Transformer(A.T.) (Xu et al., 2022), PatchTST (Nie et al., 2023), HBOS (Goldstein & Dengel, 2012), IForest (Liu et al., 2008), and PCA (Shyu et al., 2003).

**Settings.** During pre-training, HORAI is optimized using the Adam optimizer with an initial learning rate of 0.0005 and trained for 20 epochs, employing an early stopping strategy with a patience of 5 epochs. For the forecasting task, all methods predict future values at four horizons to ensure a fair comparison. Additionally, *none of the models employ the drop-last strategy* (Qiu et al., 2024). For the anomaly detection task, evaluation is conducted using three score-based metrics: AUC-ROC, VUS-ROC, and VUS-PR (Paparrizos et al., 2022a), which are threshold-independent. Notably, **time series foundation models perform zero-shot inference directly, whereas time-series-specific models are trained in a full-shot setting for comparison**.

### 4.2 TIME SERIES FORECASTING

As shown in Table 1, HORAI achieves state-of-the-art forecasting performance compared to both unimodal foundation models and multimodal time-series-specific models, achieving top performance on 14 out of 18 cases. Specifically, relative to unimodal foundation models, HORAI reduces the MSE of Sunidal by 16.9% , and outperforms ROSE with reductions of 27.2% in MSE. These results indicate that HORAI effectively leverages multimodal information to enhance time series understanding and improve predictive accuracy. Compared to multimodal time-series-specific models trained in a full-shot manner, HORAI achieves superior performance even in the zero-shot setting: exceeding GPT4MTS by 7.5% in MSE , and surpassing TimeVLM by 8.4% in MSE. This demonstrates that pre-training on the large-scale multimodal time series dataset equips HORAI with strong generalization ability for time series forecasting.

Table 1: Time series forecasting results under zero-shot and full-shot settings, reported as the average across four prediction horizons. The best results are highlighted in **bold**, and the second-best results are underlined. Full results are presented in the Table 11.

| Type | Time Series Foundation Models (Zero-Shot) | | | | | | | | | | | | Time-Series-Specific Models (Full-Shot) | | | | | | | |
|---|---|---|---|---|---|---|---|---|---|---|---|---|---|---|---|---|---|---|---|---|
| Models | HORAI | | ChatTime | | VisionTS | | ROSE | | Timer | | MOIRAI | | GPT4MTS | | TATS | | GPT4TS | | TimeVLM | |
| Metric | MSE | MAE | MSE | MAE | MSE | MAE | MSE | MAE | MSE | MAE | MSE | MAE | MSE | MAE | MSE | MAE | MSE | MAE | MSE | MAE |
| Agriculture | 0.236 | 0.332 | 0.369 | 0.410 | 0.290 | 0.336 | 0.345 | 0.372 | 0.289 | 0.339 | 0.272 | 0.403 | 0.225 | 0.298 | **0.215** | 0.301 | 0.220 | **0.294** | 0.237 | 0.302 |
| Climate | **0.867** | **0.741** | 1.860 | 1.106 | 1.307 | 0.930 | 1.475 | 0.987 | 0.888 | 0.764 | 1.921 | 1.095 | 1.182 | 0.889 | 1.180 | 0.887 | 1.184 | 0.891 | 1.195 | 0.899 |
| Energy | 0.250 | 0.358 | **0.247** | **0.352** | 0.304 | 0.420 | 0.386 | 0.479 | 0.274 | 0.359 | 0.324 | 0.417 | 0.262 | 0.380 | 0.255 | 0.368 | 0.260 | 0.376 | 0.260 | 0.374 |
| Environment | **0.307** | **0.393** | 0.395 | 0.456 | 0.354 | 0.436 | 0.392 | 0.456 | 0.351 | 0.428 | 0.351 | 0.403 | 0.323 | 0.400 | 0.319 | 0.396 | 0.322 | 0.393 | 0.319 | 0.397 |
| Social Good | **0.792** | 0.451 | 1.069 | 0.535 | 1.126 | 0.618 | 1.141 | 0.581 | 0.974 | 0.489 | 1.430 | 0.651 | 0.920 | 0.451 | 0.918 | **0.428** | 0.917 | 0.476 | 0.868 | 0.444 |
| Traffic | **0.176** | 0.293 | 0.596 | 0.610 | 0.281 | 0.407 | 0.341 | 0.451 | 0.188 | 0.290 | 0.406 | 0.468 | 0.203 | 0.261 | 0.179 | 0.238 | 0.206 | 0.266 | 0.216 | 0.319 |
| EWJ | **0.591** | **0.542** | 0.887 | 0.641 | 0.645 | 0.584 | 0.706 | 0.605 | 0.696 | 0.595 | 0.937 | 0.688 | 0.626 | 0.549 | 0.612 | 0.546 | 0.607 | 0.543 | 0.609 | 0.544 |
| KR | **0.551** | **0.448** | 0.565 | 0.455 | 0.671 | 0.522 | 0.555 | 0.480 | 0.549 | 0.463 | 0.992 | 0.629 | 0.555 | 0.450 | 0.578 | 0.449 | 0.578 | 0.448 | 0.584 | 0.454 |
| MDT | **0.373** | **0.434** | 0.496 | 0.479 | 0.433 | 0.485 | 0.461 | 0.493 | 0.389 | 0.448 | 0.606 | 0.569 | 0.385 | 0.442 | 0.389 | 0.436 | 0.391 | 0.438 | 0.392 | 0.437 |

## 4.3 TIME SERIES ANOMALY DETECTION

As illustrated in Table 2, HORAI achieves state-of-the-art anomaly detection performance compared to both unimodal foundation models and time-series-specific models, attaining top results on 13 out of 15 cases. Compared to DADA, a general time series anomaly detector, HORAI outperforms it by 14.6%, 22.4%, and 23.6% in AUC-ROC, VUS-ROC, and VUS-PR, respectively, under the zero-shot setting. This highlights that integrating multimodal data, such as text and images, enables the model to identify anomalous patterns better. Against time-series-specific anomaly detection models, HORAI outperforms GPT4TS by 13.2%, 23.8%, and 26.7% in AUC-ROC, VUS-ROC, and VUS-PR, respectively. These results demonstrate that pre-training on large-scale, multi-domain data equips HORAI with robust general detection capability, effectively distinguishing between diverse normal and anomalous patterns.

Table 2: Time series anomaly detection results under zero-shot and full-shot settings. The best results are in **bold**, and the second-best results are underlined. More metric results are in Table 12.

| Type | | Time Series Foundation Models (Zero-Shot) | | | | Time-Series-Specific Models (Full-shot) | | | | | | | | |
|---|---|---|---|---|---|---|---|---|---|---|---|---|---|---|
| Datasets | Metric | HORAI | DADA | Timer | UniTS | GPT4TS | LLMMixer | TimesNet | DCdetector | A.T. | PatchTST | HBOS | IForest | PCA |
| EWJ | AUC-ROC | **86.32** | 79.11 | 76.15 | 79.87 | 75.58 | 57.69 | 82.39 | 53.40 | 43.81 | 78.53 | 71.82 | 69.20 | 54.35 |
| | VUS-ROC | **82.13** | 71.79 | 67.72 | 73.91 | 67.95 | 52.79 | 75.76 | 47.10 | 31.75 | 71.96 | 62.07 | 59.24 | 45.26 |
| | VUS-PR | **45.89** | 43.36 | 33.17 | 39.32 | 35.63 | 15.13 | 43.15 | 15.37 | 10.85 | 36.08 | 41.19 | 37.81 | 19.38 |
| MDT | AUC-ROC | **90.74** | 79.04 | 75.65 | 73.19 | 74.79 | 60.30 | 86.67 | 53.82 | 56.44 | 84.55 | 60.26 | 63.92 | 54.51 |
| | VUS-ROC | **87.02** | 66.76 | 60.28 | 58.67 | 62.30 | 46.80 | 83.40 | 45.02 | 44.53 | 77.69 | 55.30 | 54.02 | 44.09 |
| | VUS-PR | **52.72** | 46.81 | 38.38 | 37.61 | 44.81 | 15.21 | 52.13 | 15.72 | 15.93 | 41.67 | 44.77 | 35.32 | 22.93 |
| KR | AUC-ROC | **91.41** | 79.53 | 66.72 | 80.95 | 78.30 | 65.77 | 85.88 | 52.97 | 51.25 | 82.15 | 75.16 | 74.45 | 63.58 |
| | VUS-ROC | **86.77** | 70.82 | 75.99 | 73.93 | 67.81 | 47.06 | 79.00 | 43.04 | 41.97 | 74.65 | 60.70 | | 47.51 |
| | VUS-PR | **58.58** | 45.90 | 51.41 | 43.32 | 38.23 | 19.10 | 51.60 | 8.49 | 7.94 | 36.18 | 54.17 | 43.31 | 24.19 |
| Energy | AUC-ROC | **69.53** | 62.33 | 60.54 | 63.38 | 66.54 | 61.31 | 68.36 | 48.75 | 38.68 | 66.70 | 60.80 | 60.32 | 61.14 |
| | VUS-ROC | **61.46** | 54.37 | 46.03 | 51.15 | 53.10 | 53.04 | 59.47 | 45.93 | 31.56 | 58.31 | 51.50 | 53.61 | 53.07 |
| | VUS-PR | 35.15 | 34.18 | 29.46 | 31.04 | 31.68 | 30.35 | 38.61 | 22.57 | 19.69 | 34.41 | 42.57 | **46.03** | 44.30 |
| Weather | AUC-ROC | 81.49 | 66.37 | 80.86 | 81.22 | 74.47 | 79.60 | 81.10 | 47.90 | 47.11 | 82.02 | 64.47 | 67.81 | 67.71 |
| | VUS-ROC | 80.40 | 61.03 | 73.22 | 75.08 | 70.03 | 71.71 | **81.91** | 45.56 | 43.32 | 79.97 | 54.16 | 56.45 | 57.38 |
| | VUS-PR | **50.76** | 30.00 | 43.21 | 44.35 | 41.30 | 43.47 | 50.09 | 18.33 | 19.17 | 50.13 | 46.58 | 49.66 | 47.13 |

## 4.4 ABLATION STUDY

To evaluate the effectiveness of each component in HORAI, we conduct ablation experiments. Figure 3 illustrates the unique impact of each module. Removing the image and text modalities (W/O Modality) leads to a drop in performance, demonstrating that HORAI effectively leverages textual semantics and visual spatial information to enhance time series modeling. In the Modality Exchange

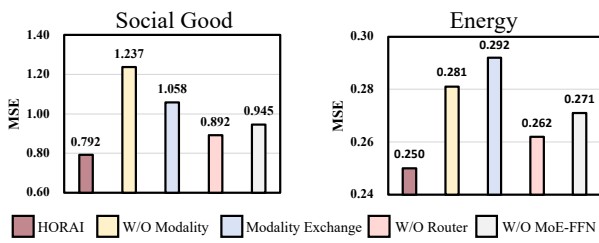

Figure 3: Ablation study on the Social Good dataset and the Energy dataset.

variant, mid- and high-frequency time series features are aligned with texts, while low-frequency features are aligned with images. In contrast, HORAI aligns low-frequency features with text and mid- to high-frequency features with images, effectively exploiting the correspondence between modality-specific information and different frequency components of the time series, which improves modeling performance. This demonstrates that frequency-aware cross-modality alignment is crucial for capturing complementary patterns across modalities. Replacing the Time-Frequency MoE-FFN with a standard FFN (W/O MoE-FFN) shows that the MoE-FFN allows each expert to capture distinct patterns, thereby enhancing the model's generalization ability. Removing frequency information from the router (W/O Router) demonstrates that incorporating frequency information helps guide multimodal tokens to the most appropriate FFN experts, further improving performance.

## 4.5 MODEL ANALYSIS

**Fine-tune with downstream data.**  To examine how the amount of fine-tuning data affects downstream performance, we evaluate HORAI by progressively enlarging the training portion of the Environment dataset. As shown in Figure 7 (a), the forecasting accuracy steadily improves as more data is used, reaching its best with the full dataset. Specifically, the MAE decreases from 0.393 to 0.370, and the MSE decreases from 0.307 to 0.259. These results highlight HORAI's strong adaptability to downstream data availability.

**Model Size Analysis.**  Scalability is a fundamental property of foundation models. To assess the scalability of HORAI, we construct different variants by varying the number of Time-Frequency Decoder layers and the model dimension $D_{model}$, and pre-train them on the proposed MM-TS dataset, followed by evaluation on the environment dataset. Specifically, in the first setting, we keep $D_{model}$ fixed and increase the number of Decoder layers from 3 to 6 and then to 12. In the second setting, we fix the number of Decoder layers while enlarging $D_{model}$ from 256 to 768 and further to 1024. As shown in Figure 7 (b), increasing the number of layers consistently enhances performance, with MSE reduced from 0.313 to 0.305. Similarly, enlarging the model dimension also leads to forecasting performance improvements, as MSE decreases from 0.330 to 0.299. These results clearly demonstrate HORAI's scalability to larger model capacities.

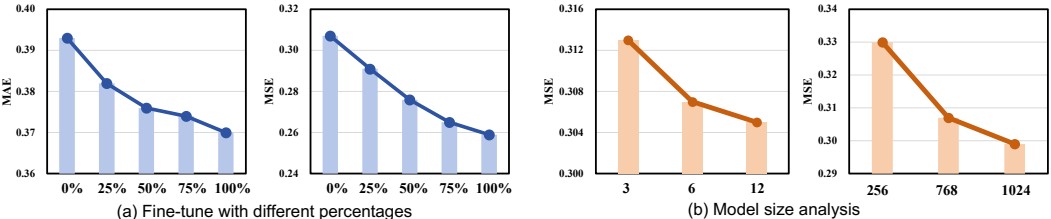

Figure 4: (a) Fine-tuning HORAI with different data percentages on the Environment dataset (b) Model performance on the different model dimensions and the number of decoder layers.

**Text Replacement.**  To examine whether HORAI truly leverages semantic information from text to enhance time series analysis, we conduct text replacement experiments with three variants: using randomly generated text (Random Text), substituting all samples with a single global domain description derived from dataset information (Domain Text), and removing the text modality altogether (W/O Text).

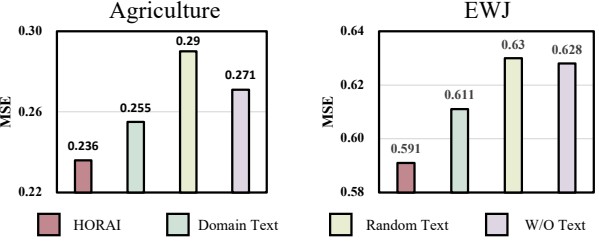

Figure 5: Text replacement experiments on the Agriculture dataset and the EWJ dataset.

As shown in Table 5, introducing random text leads to a substantial performance drop, even worse than removing the text modality, indicating that HORAI does not simply rely on the presence of text but actually understands and exploits its semantic content. Similarly, when every sample is

assigned the same global domain description, the model's performance also declines, suggesting that sample-specific semantic information is crucial for effective time series analysis.

## 5 CONCLUSION

In this paper, we take an early step toward multimodal foundation models for time series analysis. On the pre-training data side, we construct MM-TS, a large-scale multimodal dataset spanning time series, text, and image across six domains, with more than one billion time points. This dataset provides a solid foundation for studying multimodal foundation models. On the modeling side, we propose HORAI, a frequency-enhanced multimodal foundation model. It integrates two core components: the Frequency-guided Cross-Modality Encoder and the Time-Frequency Decoder, effectively fusing different multimodal features and enhancing model generalization across domains and modalities. After pre-training on MM-TS, HORAI achieves state-of-the-art performance in time series forecasting and anomaly detection tasks, which demonstrates strong task versatility and generalization ability.

ETHICS STATEMENT

Our work is conducted on publicly available benchmark datasets, without involving any additional personal information. For the construction of the MM-TS pretraining dataset, the time series modality is collected from public sources, while the textual modality is generated using large language models. No human subjects are involved in this research.

REPRODUCIBILITY STATEMENT

The performance of HORAI and datasets used in our work are real, and all experimental results can be reproduced. Once the paper is accepted, we will release the code of HORAI and pre-training dataset MM-TS.

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

# A    DATASETS

## A.1    PRE-TRAIN DATASET MM-TS

For time series modality, we assemble a large and diverse set of publicly available time series datasets covering domains such as energy, nature, transportation, web, health, and economics. The corpus contains around 1 billion time points, with a strict separation from all target evaluation datasets. The datasets vary widely in their sampling frequencies—from millisecond-level measurements to monthly observations—reflecting both the heterogeneity of real-world scenarios and the complexity of temporal dynamics.

Table 3: List of pretraining datasets of time series modality.

| Domain | Dataset | Frequency | Time Pionts | Source |
|---|---|---|---|---|
| Energy | Aus. Electricity Demand | Half Hourly | 1155264 | Monash (Godahewa et al., 2021) |
| | Wind | 4 Seconds | 7397147 | Monash (Godahewa et al., 2021) |
| | Wind Farms | Minutely | 172178060 | Monash (Godahewa et al., 2021) |
| | Solar Power | 4 Seconds | 7397222 | Monash (Godahewa et al., 2021) |
| | London Smart Meters | Half Hourly | 166527216 | Monash (Godahewa et al., 2021) |
| | BDG-2 Rat | Hourly | 4596080 | (Alexandrov et al., 2020) |
| | BDG-2 Panther | Hourly | 893840 | (Alexandrov et al., 2020) |
| | BDG-2 Fox | Hourly | 2285288 | (Alexandrov et al., 2020) |
| Nature | Phoneme | - | 2160640 | UCRDau et al. (2019) |
| | EigenWorms | - | 27947136 | UEA (Bagnall et al., 2018) |
| | PRSA | Hourly | 4628448 | (Zhang et al., 2017) |
| | Temperature Rain | Daily | 23252200 | Monash (Godahewa et al., 2021) |
| | StarLightCurves | - | 9457664 | UCR (Dau et al., 2019) |
| | Worms | 0.033 Seconds | 232200 | UCR (Dau et al., 2019) |
| | Saugeen River Flow | Daily | 23741 | Monash (Godahewa et al., 2021) |
| | Sunspot | Daily | 73924 | Monash (Godahewa et al., 2021) |
| | Weather | Daily | 43032000 | Monash (Godahewa et al., 2021) |
| | KDD Cup 2018 | Daily | 2942364 | MonashGodahewa et al. (2021) |
| | US Births | Daily | 7305 | Monash (Godahewa et al., 2021) |
| Healthcare | MotorImagery | 0.001 Seconds | 72576000 | UEA (Bagnall et al., 2018) |
| | AtrialFibrillation | 0.008 Seconds | 38400 | UEA (Bagnall et al., 2018) |
| | PigArtPressure | - | 624000 | UCR (Dau et al., 2019) |
| | PIGCVP | - | 624000 | UCR (Dau et al., 2019) |
| | TDbrain | 0.002 Seconds | 79232703 | (Wang et al., 2024) |
| Transport | Pems03 | 5 Minute | 9382464 | (Liu et al., 2022) |
| | Pems04 | 5 Minute | 5216544 | (Liu et al., 2022) |
| | Pems07 | 5 Minute | 24921792 | (Liu et al., 2022) |
| | Pems08 | 5 Minute | 3035520 | (Liu et al., 2022) |
| | Pems-bay | 5 Minute | 16937700 | (Liu et al., 2022) |
| | Pedestrian_Counts | Hourly | 3132346 | Monash (Godahewa et al., 2021) |
| | SZ-Taxi | 15 Minute | 464256 | (Wang et al., 2023) |
| | Taxi | Half Hourly | 40584636 | (Alexandrov et al., 2020) |
| | Uber TLC | Hourly | 510284 | (Alexandrov et al., 2020) |
| Web | Web Traffic | Daily | 116485589 | Monash (Godahewa et al., 2021) |
| Economic | FRED_MD | Monthly | 77896 | (McCracken & Ng, 2016) |
| | Bitcoin | Daily | 75364 | Monash (Godahewa et al., 2021) |
| | NN5 | Daily | 87801 | (Taieb et al., 2012) |

For text modality, considering that pre-training foundation models require large amounts of high-quality textual modality data, and that in real-world scenarios such text is often difficult to obtain, scarce, and may contain noise or irrelevant information, we design specific prompts and leverage large language models to generate large-scale, high-quality textual data. Taking the London Smart Meters dataset as an example, we construct the following prompt:

**Prompt:** You are a domain expert in {**energy domain**} systems and time series analysis, tasked with generating a detailed yet concise textual summary of time series data. The provided input is a univariate time series: {**time series data**}, sourced from the {**London Smart Meters**} dataset within the {**energy domain**}, with observations collected at regular half-hour intervals from {**start time**} to {**end time**}. Consider the broader contextual factors affecting this dataset, including seasonal variations, regional energy usage patterns, socio-economic events, and policy changes during the given period. Analyze the temporal progression of the data and summarize the key trends in a single coherent paragraph. Focus on identifying and describing patterns such as upward or downward trends, stable periods, sudden spikes or drops, cyclic behaviors, anomalies, and general fluctuations. Your description begin with: "The {London Smart Meters} series exhibits..."Ensure your summary integrates both statistical patterns and contextual reasoning, presenting a holistic overview of how the values evolve over time. Use precise, objective, and professional language.

Figure 6: The prompt designed for generating textual descriptions of the London Smart Meters dataset.

## A.2    EVALUATION DATASET

To evaluate HORAI in a multi-task setting, we employ widely used benchmark datasets for both forecasting and anomaly detection. 1) Forecasting: As shown in Table 4, experiments are conducted on TimeMMD (Liu et al., 2024a) and additional datasets (Dong et al., 2024), covering diverse domains such as Agriculture, Climate, Energy, Environment, Social Good, Traffic, EWJ, KR, and MDT. 2) Anomaly Detection: We evaluate HORAI on five datasets—Weather, Energy, KR, EWJ, and MDT—with anomaly ratios ranging from 5.81% to 17.23%. Detailed statistics are provided in Table 5.

Table 4: The statistics of evaluation datasets for the forecasting task.

| Tasks | Dataset | Variate | Frequency | Dataset Size | Timespan |
|---|---|---|---|---|---|
| Forecasting | Agriculture | 1 | Monthly | 496 | 1983-2024 |
| | Climate | 5 | Monthly | 496 | 1983-2024 |
| | Energy | 9 | Weekly | 1479 | 1996-2024 |
| | Environment | 4 | Daily | 11102 | 1982-2023 |
| | Social Good | 1 | Monthly | 900 | 1950-2024 |
| | Traffic | 1 | Monthly | 531 | 1980-2024 |
| | EWJ | 1 | Daily | 2658 | 2009-2020 |
| | KR | 1 | Daily | 2655 | 2009-2020 |
| | MDT | 1 | Daily | 2732 | 2009-2020 |

Table 5: The statistics of evaluation datasets for the anomaly detection task.

| Tasks | Dataset | Anomaly Ratio | Frequency | Dataset Description |
|---|---|---|---|---|
| Detection | Weather | 17.10% | Monthly | Temperature and humidity information collected from government websites. |
| | Energy | 17.23% | Weekly | The dataset records weekly U.S. gasoline prices (dollars per gallon). |
| | KR | 6.21% | Daily | The dataset is collected from Yahoo, NASDAQ finance websites. |
| | MDT | 11.17% | Monthly | The dataset is collected from Yahoo, NASDAQ finance websites. |
| | EWJ | 9.96% | Daily | The dataset is collected from Yahoo, NASDAQ finance websites. |

## B    BASELINES

We categorize the baselines into three groups: *Unimodal Time Series Foundation Models*, *Multi-modal Time-Series-Specific Models*, and *Unimodal Time-Series-Specific Models*. Unimodal Time Series Foundation Models are pre-trained on large-scale, cross-domain unimodal time series data,

enabling direct inference on downstream tasks and demonstrating certain generalization capabilities. In contrast, Time-Series-Specific Models require training on each downstream dataset and can be further divided based on the input type. Multimodal Time-Series-Specific Models leverage additional modalities, such as text or images, or reuse LLM representations to enhance time series understanding. Unimodal Time-Series-Specific Models, on the other hand, design tailored modules to exploit the inherent characteristics of time series data.

### B.1 UNIMODAL TIME SERIES FOUNDATION MODELS

- Sundial (Liu et al., 2025) proposes a TimeFlow Loss that predicts the distribution of the next patch, enabling Transformer training without discrete tokenization and supporting probabilistic forecasting.
- VisionTS (Chen et al., 2024a) converts time series data into image form and uses visual mask autoencoders for unsupervised feature learning.
- ROSE (Wang et al., 2025b) combines frequency decomposition with time-series registers to jointly learn both domain-invariant and domain-specific representations, facilitating knowledge transfer to downstream tasks.
- Timer (Liu et al., 2024c) adopts a decoder-only architecture employing autoregressive modeling for generative pre-training.
- MOIRAI (Woo et al., 2024) introduces multi-scale patch projections to model diverse patterns and an any-variate attention mechanism that allows flexible handling of time series with arbitrary dimensionality.
- DADA (Shentu et al., 2025) leverages adaptive bottleneck and dual-adversarial decoding to enable robust zero-shot anomaly detection across diverse domains.
- UniTS (Gao et al., 2024) proposes a novel unified network backbone for classification, forecasting, and anomaly detection.

### B.2 MULTIMODAL TIME-SERIES-SPECIFIC MODELS

- GPT4MTS (Jia et al., 2024) propose a prompt tuning-based LLM for time series forecasting with multimodal input.
- TATS (Li et al., 2025) propose a plug-and-play multimodal time series forecasting framework, which transforms text representations into auxiliary variables.
- GPT4TS (Zhou et al., 2023) fine-tunes the limited parameters of LLM, demonstrating competitive performance by transferring knowledge from large-scale pre-training text data.
- LLMMixer (Kowsher et al., 2024) adapts LLMs for time series forecasting by breaking down the data into different time scales.
- TimeVLM (Zhong et al., 2025) leverages pre-trained VLMs to enhance time series forecasting by unifying temporal, visual, and textual information.

### B.3 UNIMODAL TIME-SERIES-SEPCIFIC MODELS

- TimesNet (Wu et al., 2023) transforms the 1D time series into a set of 2D tensors based on multiple periods to handle the multi-periodicity of the time series.
- DCdetector (Yang et al., 2023) leverages dual-attention contrastive representation learning, extracting normal feature representations through self-supervised learning and dual-attention mechanisms.
- Anomaly Transformer (Xu et al., 2022) leverages a self-attention mechanism to capture both short- and long-term dependencies in time series, and detects anomalies by analyzing differences in association matrices.
- PatchTST (Nie et al., 2023) segments time series into subseries-level patches that serve as input tokens to the Transformer and applies the channel-independence strategy for training on multivariate time series.
- HBOS (Goldstein & Dengel, 2012) is a fast unsupervised anomaly detection method based on histogram density estimation.

- IForest (Liu et al., 2008) detects anomalies by recursively partitioning data to isolate out-liers, rather than modeling normal behavior.
- PCA (Shyu et al., 2003) detects anomalies by measuring deviations in the principal com-ponent space, assuming outliers lie far from the normal distribution.

## C EXPERIMENT SETTING

*During pre-training*, HORAI is optimized using the Adam optimizer with an initial learning rate of 0.0005 and trained for 20 epochs, with early stopping applied using a patience of 10 epochs. The batch size is set to 2048, the input time series length to 576, and the patch size to 48. The Time-Frequency Decoder is configured with 6 layers, the model dimension $D_{model}$ is set to 768, and the ratio parameter $\alpha$ for high- and low-frequency decomposition is fixed at 0.05. All experiments are implemented in PyTorch, and pre-training is conducted on four NVIDIA Tesla A800 80GB GPUs.

*For forecasting*, To ensure fairness, we remove the drop-last strategy for HORAI and all base-lines, since using it would result in inconsistent numbers of test samples across different batch sizes Qiu et al. (2024). For each dataset, we evaluate four prediction horizons for both HORAI and the baselines. Specifically, Agriculture, Climate, Social Good, Traffic, EWJ, KR, and MDT are evaluated with horizons $\{6, 8, 10, 12\}$, Environment with $\{48, 96, 192, 336\}$, and Energy with $\{12, 24, 36, 48\}$.

## D EFFICIENCY ANALYSIS

We compare HORAI with unimodal foundation models and multimodal end-to-end models using three common efficiency metrics: the number of parameters, MACs, and inference time. All ex-periments are conducted on the Environment dataset with a batch size of 1. As shown in Table 6, HORAI does not exhibit a significant efficiency disadvantage compared to unimodal foundation models, despite incorporating multimodal information. Specifically, HORAI achieves lower MACs than VisionTS and MOIRAI, and demonstrates faster inference speed compared to Sundial, while also delivering superior prediction performance on multimodal time series datasets. When compared with multimodal time-series-specific models, although these models generally have few parameters, low MACs, and short inference times, they require retraining on each downstream dataset. In con-trast, HORAI supports direct zero-shot inference, which makes it far more efficient in terms of overall time cost.

Table 6: Efficiency analysis on the environment dataset.

| Models | Parameters(M) | MACs | Inference(s) |
|---|---|---|---|
| TimeVLM | 152 | 2.24 G | 0.0576 |
| GPT4MTS | 167 | 1.21 G | 0.0611 |
| GPT4TS | 85 | 514.36 M | 0.0272 |
| TaTS | 83 | 14.77 M | 0.0686 |
| Sundial | 128 | 1.32 G | **0.0813** |
| VisionTS | 112 | **5.51** G | 0.0073 |
| ROSE | 16 | 85.41 M | 0.0542 |
| Timer | 84 | 84.14 M | 0.0048 |
| MOIRAI | 310 | 4.23 G | 0.0511 |
| HORAI | **426** | 3.51 G | 0.0733 |

## E DISCUSSION

We further provide a clarified discussion comparing ChatTime and HORAI, including the following aspects: (1) **Model perspective**: HoRAI is **specifically architected as a multimodal foundation model** integrating time series, images, and text. It leverages modality-specific encoders to extract

distinct features and employs a novel frequency-enhanced alignment to explicitly fuse these representations from multiple perspectives. In contrast, ChatTime **adapts general-purpose LLMs** for time series analysis. While leveraging LLMs' inherent reasoning abilities for time series analysis offers generalization, discretizing continuous numerical values into textual tokens leads to precision loss, making it difficult to capture time series patterns. (2) **Data perspective**: HORAI is pretrained on a large-scale multimodal dataset incorporating aligned text and images. These modalities capture diverse characterizations of temporal dynamics from multiple perspectives and simultaneously introduce some external context, providing relevant supervision that improves generalization. However, ChatTime relies only on simple prompts such as "Please predict the following sequence," which offer limited text regarding the specific time series characteristics.

# F  MODEL ANALYSIS

## F.1  SENSITIVITY ANALYSIS

We conduct sensitivity experiments on two key parameters: the frequency threshold $\alpha$ and the number of selected experts $K$. As shown in the Table 7, setting $\alpha$ to 0.05 achieves the best prediction performance. This value distinctly partitions low-frequency from mid-to-high-frequency features, facilitating optimal alignment with text and image modalities. Conversely, a larger $\alpha$ forces excessive information into high-frequency components, thereby amplifying noise-like patterns; whereas an overly small $\alpha$ introduces redundant low-frequency information, which disrupts the alignment between image and time series representations. As shown in Table 8, selecting the Top-2 or Top-3 experts yields superior performance. Activating all experts tends to introduce redundancy from irrelevant experts, thereby diluting the model's generalization. Whereas selecting only a single expert limits the representational capacity, preventing the model from modeling diverse time series patterns.

Table 7: Hyper-parameter sensitivity analysis about the frequency threshold $\alpha$.

| Metrics | $\alpha = 0.01$ MSE | $\alpha = 0.05$ MSE | $\alpha = 0.25$ MSE | $\alpha = 0.5$ MSE |
|---|---|---|---|---|
| Agriculture | 0.245 | **0.236** | 0.255 | 0.277 |
| Climate | 0.868 | **0.867** | 1.054 | 1.200 |
| Energy | 0.260 | **0.250** | 0.342 | 0.335 |
| Environment | 0.313 | **0.307** | 0.332 | 0.333 |

Table 8: Hyper-parameter sensitivity analysis about the number of selected experts $K$.

| Metrics | K=1 MSE | K=2 MSE | K=3 MSE | K=4 MSE |
|---|---|---|---|---|
| Agriculture | 0.258 | 0.236 | **0.232** | 0.252 |
| Climate | 1.062 | **0.867** | 0.884 | 0.896 |
| Energy | 0.262 | **0.250** | 0.260 | 0.265 |
| Environment | 0.320 | **0.307** | 0.315 | 0.326 |

## F.2  ABLATION ANALYSIS ON SPECIFIC MODALITIES AND ALIGNMENT STRATEGIES

We perform ablation studies to evaluate the contributions of individual modalities (text, image) and the efficacy of our frequency-based alignment strategy. Specifically, we analyze four settings: 1) only text and time series; 2) only image and time series ; 3) text, image, and time series without frequency-based alignment (w/o Freq-Align); and 4) swapping modalities by fusing low-frequency time series with images and mid-to-high frequency time series with text (Modality Exchange). As

shown in Table 9, both visual and textual modalities contribute to performance gains, though their relative impact varies depending on the dataset characteristics. For datasets exhibiting clear long-term trends, such as Agriculture and Energy, the text modality contributes more significantly. Conversely, for datasets dominated by local fluctuations, such as Climate, the image modality proves more critical. Crucially, the significant performance drop observed when removing frequency-based alignment and modality exchange underscores the validity of our design: it confirms that aligning images with mid-to-high frequency components and text with low-frequency components is the most effective strategy.

Table 9: Ablation analysis about each modality.

| Metrics | HORAI | | Text + Time Series | | Image + Time Series | | W/O Freq-Align | | Modality Exchange | |
| --- | --- | --- | --- | --- | --- | --- | --- | --- | --- | --- |
| | MSE | MAE | MSE | MAE | MSE | MAE | MSE | MAE | MSE | MAE |
| Agriculture | **0.236** | **0.332** | 0.271 | 0.349 | 0.312 | 0.360 | 0.266 | 0.347 | 0.292 | 0.352 |
| Climate | **0.867** | **0.741** | 1.102 | 0.828 | 0.982 | 0.797 | 0.928 | 0.786 | 1.321 | 0.856 |
| Energy | **0.250** | **0.358** | 0.295 | 0.405 | 0.306 | 0.415 | 0.290 | 0.398 | 0.292 | 0.402 |
| Environment | **0.307** | **0.393** | 0.344 | 0.412 | 0.320 | 0.395 | 0.325 | 0.396 | 0.360 | 0.426 |

### F.3 ABLATION ANALYSIS ABOUT TEXT ENCODER AND VISION ENCODER

To evaluate the model's performance with different encoders, we conduct additional experiments by replacing both text and visual encoders. Considering time and computational constraints, we select encoders with relatively small parameter sizes. Specifically, the text encoders include GPT2-large, LLaMA3-1B, and Qwen2.5-1.5B, while the visual encoder comparison uses Swin Transformer. As shown in the Table 10, for a given text encoder, models with larger parameter sizes tend to perform slightly better, and employing more advanced architectures (e.g., Qwen and LLaMA) generally yields further improvements. In the comparison of visual encoders, ViT and Swin Transformer achieve similar overall forecasting performance.

Table 10: Ablation analysis of different text encoders and image encoders.

| Metrics | HORAI | | GPT2-Large | | Llama3-1B | | Qwen-1.5B | | Swin Transformer-Base | |
| --- | --- | --- | --- | --- | --- | --- | --- | --- | --- | --- |
| | MSE | MAE | MSE | MAE | MSE | MAE | MSE | MAE | MSE | MAE |
| Agriculture | **0.236** | **0.332** | 0.258 | 0.352 | 0.228 | 0.315 | 0.237 | 0.334 | 0.230 | 0.325 |
| Climate | 0.867 | 0.741 | 0.913 | 0.842 | 0.874 | 0.752 | **0.850** | **0.732** | 0.876 | 0.761 |
| Energy | 0.250 | 0.358 | 0.265 | 0.372 | 0.245 | 0.350 | **0.229** | **0.342** | 0.254 | 0.364 |
| Environment | 0.307 | 0.393 | 0.325 | 0.398 | 0.310 | 0.398 | **0.304** | **0.392** | 0.300 | 0.388 |

### F.4 CONVERGENCE ANALYSIS OF FULL-SHOT TIME-SERIES-SPECIFIC MODELS

### F.5 VISUALIZATION ANALYSIS OF FREQUENCY-BASED ALIGNMENT

To visually verify the efficacy of the frequency-based alignment, we employ t-SNE to visualize the learned embeddings of the Energy dataset. As shown in Figure 8 of the revised paper, distinct alignment patterns emerge: Image features align closely with Mid-to-High Frequency time series components, while Text features cluster tightly with Low-Frequency components. This visual evidence empirically confirms the effectiveness of our frequency-guided alignment mechanism.

## G THE USE OF LARGE LANGUAGE MODELS (LLMS)

In our proposed method, HORAI, we employ LLMs as the text tokenizer and text encoder to extract semantic features and fuse with time series and image, enhancing the model's ability for time series understanding. For the constructed pre-training dataset MM-TS, we leverage DeepSeek to generate

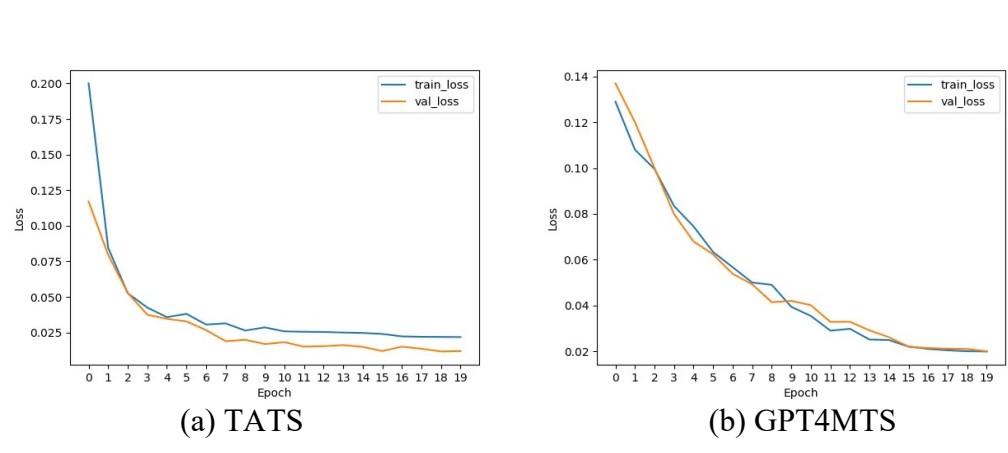

(a) TATS                                    (b) GPT4MTS

Figure 7: Training and validation losses of TATS and GPT4MTS on the Traffic dataset.

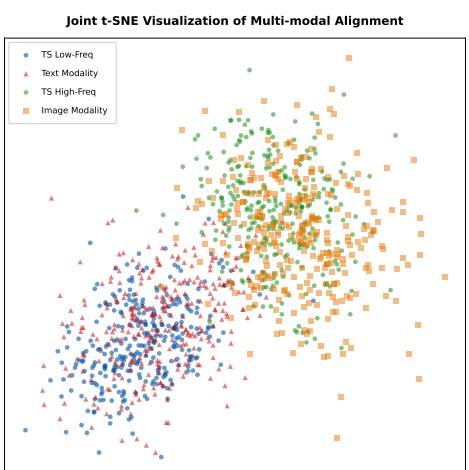

Figure 8: T-SNE visualization of frequency-based alignment.

Table 11: Full time series forecasting results of HORAI, time series foundation models, and time-series-specific models.

| Type | | Time Series Foundation Models (Zero-Shot) | | | | | | | | | | | | Time-Series-Specific-Models (Full-Shot) | | | | | | | |
|---|---|---|---|---|---|---|---|---|---|---|---|---|---|---|---|---|---|---|---|---|---|
| Models | | HORAI | | ChatTime | | VisionTS | | ROSE | | Timer | | MOIRAI | | GPT4MTS | | TATS | | GPT4TS | | TimeVLM | |
| Metric | | MSE | MAE | MSE | MAE | MSE | MAE | MSE | MAE | MSE | MAE | MSE | MAE | MSE | MAE | MSE | MAE | MSE | MAE | MSE | MAE |
| Agriculture | 6 | 0.150 | 0.274 | 0.243 | 0.340 | 0.210 | 0.289 | 0.219 | 0.299 | 0.168 | 0.272 | 0.187 | 0.342 | 0.161 | 0.257 | 0.140 | 0.251 | **0.135** | **0.242** | 0.143 | 0.245 |
| | 8 | 0.209 | 0.318 | 0.349 | 0.399 | 0.266 | 0.323 | 0.278 | 0.339 | 0.243 | 0.317 | 0.245 | 0.391 | 0.207 | 0.288 | **0.187** | **0.282** | 0.198 | 0.284 | 0.215 | 0.287 |
| | 10 | 0.273 | 0.355 | 0.390 | 0.418 | 0.307 | 0.348 | 0.408 | 0.406 | 0.328 | 0.361 | 0.297 | 0.423 | 0.230 | 0.305 | **0.244** | 0.320 | 0.258 | **0.313** | 0.271 | 0.320 |
| | 12 | 0.312 | 0.382 | 0.497 | 0.483 | 0.376 | 0.386 | 0.474 | 0.443 | 0.415 | 0.405 | 0.357 | 0.455 | 0.301 | 0.342 | **0.290** | 0.350 | 0.291 | **0.338** | 0.322 | 0.359 |
| | avg | 0.236 | 0.332 | 0.369 | 0.410 | 0.290 | 0.336 | 0.345 | 0.372 | 0.289 | 0.339 | 0.272 | 0.403 | 0.225 | 0.298 | **0.215** | 0.301 | 0.220 | **0.294** | 0.237 | 0.302 |
| Climate | 6 | **0.846** | **0.731** | 1.884 | 1.118 | 1.316 | 0.932 | 1.488 | 0.993 | 0.876 | 0.759 | 1.624 | 1.016 | 1.199 | 0.895 | 1.194 | 0.897 | 1.207 | 0.901 | 1.218 | 0.907 |
| | 8 | **0.861** | **0.740** | 1.843 | 1.100 | 1.312 | 0.935 | 1.598 | 1.031 | 0.885 | 0.763 | 2.148 | 1.152 | 1.205 | 0.899 | 1.178 | 0.886 | 1.191 | 0.892 | 1.181 | 0.914 |
| | 10 | **0.875** | **0.746** | 1.806 | 1.090 | 1.302 | 0.928 | 1.401 | 0.967 | 0.893 | 0.766 | 1.983 | 1.112 | 1.173 | 0.885 | 1.170 | 0.881 | 1.169 | 0.886 | 1.179 | 0.880 |
| | 12 | **0.887** | **0.748** | 1.909 | 1.117 | 1.297 | 0.925 | 1.414 | 0.957 | 0.899 | 0.770 | 1.929 | 1.101 | 1.152 | 0.876 | 1.179 | 0.885 | 1.171 | 0.883 | 1.203 | 0.896 |
| | avg | **0.867** | **0.741** | 1.860 | 1.106 | 1.307 | 0.930 | 1.475 | 0.987 | 0.888 | 0.764 | 1.921 | 1.095 | 1.182 | 0.887 | 1.180 | 0.887 | 1.184 | 0.891 | 1.195 | 0.899 |
| Energy | 12 | 0.108 | 0.233 | **0.104** | **0.222** | 0.173 | 0.313 | 0.268 | 0.401 | 0.118 | 0.236 | 0.183 | 0.309 | 0.111 | 0.244 | 0.105 | 0.232 | 0.111 | 0.243 | 0.114 | 0.253 |
| | 24 | 0.211 | 0.332 | **0.203** | **0.321** | 0.264 | 0.395 | 0.363 | 0.469 | 0.225 | 0.336 | 0.290 | 0.396 | 0.232 | 0.362 | 0.216 | 0.344 | 0.223 | 0.355 | 0.227 | 0.359 |
| | 36 | 0.299 | 0.404 | **0.292** | **0.396** | 0.346 | 0.454 | 0.413 | 0.497 | 0.328 | 0.403 | 0.367 | 0.449 | 0.308 | 0.418 | 0.309 | 0.418 | 0.314 | 0.423 | 0.309 | 0.410 |
| | 48 | **0.381** | 0.466 | 0.389 | 0.470 | 0.434 | 0.516 | 0.501 | 0.549 | 0.424 | **0.460** | 0.457 | 0.515 | 0.398 | 0.496 | 0.391 | 0.480 | 0.393 | 0.484 | 0.390 | 0.475 |
| | avg | 0.250 | 0.358 | **0.247** | **0.352** | 0.304 | 0.420 | 0.386 | 0.479 | 0.274 | 0.359 | 0.324 | 0.417 | 0.262 | 0.380 | 0.255 | 0.368 | 0.260 | 0.376 | 0.260 | 0.374 |
| Environment | 48 | **0.300** | **0.385** | 0.343 | 0.406 | 0.345 | 0.426 | 0.402 | 0.459 | 0.358 | 0.431 | 0.352 | 0.404 | 0.315 | 0.400 | 0.307 | 0.389 | 0.320 | 0.396 | 0.304 | 0.387 |
| | 96 | **0.317** | **0.399** | 0.369 | 0.465 | 0.370 | 0.441 | 0.409 | 0.465 | 0.368 | 0.436 | 0.370 | 0.415 | 0.340 | 0.401 | 0.334 | 0.402 | 0.340 | 0.401 | 0.327 | 0.405 |
| | 192 | **0.307** | **0.399** | 0.377 | 0.474 | 0.360 | 0.442 | 0.389 | 0.452 | 0.351 | 0.427 | 0.350 | 0.402 | 0.336 | 0.411 | 0.332 | 0.401 | 0.330 | 0.391 | 0.328 | 0.403 |
| | 336 | 0.305 | **0.389** | 0.372 | 0.478 | 0.340 | 0.436 | 0.390 | 0.447 | 0.326 | 0.418 | 0.332 | 0.390 | 0.299 | 0.390 | 0.302 | 0.391 | **0.300** | **0.383** | 0.320 | 0.395 |
| | avg | **0.307** | **0.393** | 0.359 | 0.456 | 0.354 | 0.436 | 0.392 | 0.456 | 0.351 | 0.428 | 0.351 | 0.403 | 0.323 | 0.400 | 0.319 | 0.396 | 0.322 | 0.393 | 0.319 | 0.397 |
| Social Good | 6 | **0.660** | 0.390 | 0.988 | 0.451 | 0.957 | 0.543 | 0.939 | 0.499 | 0.845 | 0.416 | 0.966 | 0.522 | 0.718 | 0.382 | 0.753 | **0.370** | 0.717 | 0.374 | 0.732 | 0.379 |
| | 8 | **0.756** | 0.435 | 1.044 | 0.488 | 1.106 | 0.605 | 1.168 | 0.588 | 0.938 | 0.469 | 1.532 | 0.653 | 0.942 | 0.505 | 0.875 | **0.409** | 0.855 | 0.459 | 0.822 | 0.427 |
| | 10 | **0.817** | 0.470 | 1.098 | 0.519 | 1.164 | 0.636 | 1.187 | 0.595 | 1.018 | 0.515 | 1.551 | 0.691 | 0.929 | **0.446** | 0.991 | 0.459 | 0.930 | 0.463 | 0.916 | 0.465 |
| | 12 | **0.915** | 0.511 | 1.149 | 0.554 | 1.278 | 0.688 | 1.272 | 0.642 | 1.094 | 0.557 | 1.671 | 0.736 | 1.093 | **0.470** | 1.053 | 0.474 | 1.167 | 0.608 | 1.005 | 0.505 |
| | avg | **0.792** | 0.451 | 1.069 | 0.503 | 1.126 | 0.618 | 1.141 | 0.581 | 0.974 | 0.489 | 1.430 | 0.651 | 0.920 | 0.451 | 0.918 | **0.428** | 0.917 | 0.476 | 0.868 | 0.444 |
| Traffic | 6 | 0.178 | 0.297 | 0.609 | 0.623 | 0.275 | 0.411 | 0.331 | 0.449 | 0.167 | 0.267 | 0.349 | 0.448 | 0.192 | 0.264 | **0.164** | **0.226** | 0.199 | 0.278 | 0.210 | 0.316 |
| | 8 | 0.181 | 0.297 | 0.626 | 0.636 | 0.282 | 0.410 | 0.365 | 0.455 | 0.185 | 0.287 | 0.461 | 0.499 | 0.195 | 0.256 | **0.178** | **0.242** | 0.204 | 0.262 | 0.212 | 0.313 |
| | 10 | **0.175** | 0.292 | 0.572 | 0.592 | 0.286 | 0.406 | 0.326 | 0.443 | 0.196 | 0.299 | 0.414 | 0.466 | 0.204 | 0.257 | 0.185 | 0.243 | 0.210 | 0.264 | 0.222 | 0.328 |
| | 12 | **0.173** | **0.287** | 0.579 | 0.592 | 0.282 | 0.402 | 0.342 | 0.458 | 0.202 | 0.307 | 0.400 | 0.458 | 0.218 | 0.268 | 0.189 | 0.242 | 0.211 | 0.260 | 0.222 | 0.322 |
| | avg | **0.176** | 0.293 | 0.596 | 0.610 | 0.281 | 0.407 | 0.341 | 0.451 | 0.188 | 0.290 | 0.406 | 0.468 | 0.203 | 0.261 | **0.179** | **0.238** | 0.206 | 0.266 | 0.216 | 0.319 |
| EWJ | 6 | 0.555 | 0.528 | 0.808 | 0.612 | 0.583 | 0.560 | 0.634 | 0.581 | 0.643 | 0.573 | 0.751 | 0.623 | 0.579 | 0.531 | 0.550 | 0.525 | **0.550** | 0.523 | 0.552 | **0.521** |
| | 8 | **0.581** | **0.537** | 0.880 | 0.641 | 0.629 | 0.580 | 0.729 | 0.626 | 0.685 | 0.591 | 1.017 | 0.714 | 0.608 | 0.540 | 0.611 | 0.544 | 0.597 | 0.538 | 0.599 | 0.541 |
| | 10 | **0.604** | **0.550** | 0.920 | 0.652 | 0.665 | 0.591 | 0.716 | 0.599 | 0.716 | 0.604 | 0.982 | 0.705 | 0.644 | 0.559 | 0.627 | 0.551 | 0.632 | 0.551 | 0.629 | 0.554 |
| | 12 | **0.623** | **0.556** | 0.940 | 0.659 | 0.701 | 0.607 | 0.746 | 0.613 | 0.740 | 0.614 | 0.997 | 0.709 | 0.673 | 0.566 | 0.661 | 0.563 | 0.649 | 0.560 | 0.657 | 0.562 |
| | avg | **0.591** | **0.542** | 0.887 | 0.641 | 0.645 | 0.584 | 0.706 | 0.605 | 0.696 | 0.595 | 0.937 | 0.688 | 0.626 | 0.549 | 0.612 | 0.546 | 0.607 | 0.543 | 0.609 | 0.544 |
| KR | 6 | **0.533** | **0.435** | 0.528 | 0.436 | 0.628 | 0.503 | 0.687 | 0.521 | 0.530 | 0.453 | 0.793 | 0.567 | 0.528 | 0.442 | 0.542 | **0.426** | 0.539 | 0.435 | 0.550 | 0.437 |
| | 8 | **0.549** | **0.446** | 0.564 | 0.452 | 0.674 | 0.524 | 0.798 | 0.572 | 0.547 | 0.461 | 1.077 | 0.650 | 0.564 | 0.452 | 0.569 | 0.446 | 0.573 | 0.444 | 0.580 | 0.451 |
| | 10 | 0.561 | **0.454** | 0.570 | 0.459 | 0.685 | 0.526 | 0.727 | 0.530 | **0.559** | 0.468 | 1.063 | 0.649 | 0.566 | 0.455 | 0.600 | 0.462 | 0.594 | 0.452 | 0.601 | 0.463 |
| | 12 | 0.562 | **0.458** | 0.598 | 0.473 | 0.698 | 0.535 | 0.750 | 0.547 | 0.560 | 0.472 | 1.038 | 0.649 | 0.562 | 0.453 | 0.602 | 0.461 | 0.604 | 0.459 | 0.614 | 0.467 |
| | avg | **0.551** | **0.448** | 0.565 | 0.455 | 0.671 | 0.522 | 0.741 | 0.542 | 0.549 | 0.463 | 0.992 | 0.629 | 0.555 | 0.450 | 0.578 | 0.449 | 0.578 | 0.448 | 0.584 | 0.454 |
| MDT | 6 | **0.360** | 0.425 | 0.466 | 0.455 | 0.412 | 0.471 | 0.426 | 0.476 | 0.366 | 0.437 | 0.494 | 0.521 | 0.369 | 0.436 | 0.365 | 0.423 | 0.373 | **0.422** | 0.369 | 0.423 |
| | 8 | **0.369** | **0.432** | 0.474 | 0.473 | 0.431 | 0.486 | 0.483 | 0.514 | 0.383 | 0.446 | 0.668 | 0.591 | 0.377 | 0.439 | 0.383 | 0.433 | 0.386 | 0.432 | 0.385 | 0.434 |
| | 10 | **0.379** | **0.438** | 0.526 | 0.494 | 0.437 | 0.487 | 0.456 | 0.486 | 0.397 | 0.453 | 0.630 | 0.580 | 0.389 | 0.444 | 0.397 | 0.440 | 0.395 | 0.448 | 0.400 | 0.443 |
| | 12 | **0.387** | **0.442** | 0.518 | 0.494 | 0.453 | 0.495 | 0.477 | 0.499 | 0.408 | 0.458 | 0.632 | 0.582 | 0.405 | 0.450 | 0.411 | 0.447 | 0.411 | 0.452 | 0.414 | 0.448 |
| | avg | **0.373** | **0.434** | 0.496 | 0.479 | 0.433 | 0.485 | 0.461 | 0.493 | 0.389 | 0.448 | 0.606 | 0.569 | 0.385 | 0.442 | 0.389 | 0.436 | 0.391 | 0.438 | 0.392 | 0.437 |

text descriptions of time series. It is important to note that LLMs are not used for any part of the manuscript writing process.

Table 12: Time series anomaly detection results under zero-shot and full-shot settings with multiple metrics. The best results are in **bold**, and the second-best results are underlined.

| Type | | Time Series Foundation Models (Zero-Shot) | | | | Time-Series-Specific-Models (Full-Shot) | | | | | | | | |
|------|--------|-------|------|------|------|-------|---------|---------|-----------|------|---------|------|---------|------|
| Datasets | Metric | HORAI | DADA | Timer | UniTS | GPT4TS | LLMMixer | TimesNet | DCdetector | A.T. | PatchTST | HBOS | IForest | PCA |
| EWJ | Aff-F1 | **82.54** | 81.26 | 78.06 | 77.61 | 76.65 | 66.86 | 81.82 | 48.10 | 59.03 | 75.82 | 71.03 | 67.55 | 51.06 |
| | F1 | **56.28** | 49.33 | 41.21 | 39.18 | 48.33 | 18.95 | 49.37 | 17.09 | 14.39 | 45.39 | 44.80 | 41.67 | 18.68 |
| | Range-AUC-ROC | **79.84** | 68.44 | 64.71 | 67.15 | 55.57 | 41.75 | 74.22 | 45.69 | 27.50 | 69.26 | 61.02 | 57.94 | 43.78 |
| | Range-AUC-PR | **43.33** | 41.61 | 30.67 | 44.31 | 32.83 | 12.36 | 41.84 | 12.52 | 9.01 | 33.37 | 46.37 | 34.77 | 16.49 |
| | AUC-PR | 51.97 | **55.24** | 44.01 | 50.33 | 46.75 | 18.81 | 54.99 | 10.88 | 8.97 | 47.91 | 25.24 | 24.16 | 10.99 |
| | AUC-ROC | **86.32** | 79.11 | 76.15 | 79.87 | 75.58 | 57.69 | 82.39 | 53.40 | 43.81 | 78.53 | 71.82 | 69.20 | 54.35 |
| | VUS-ROC | **82.13** | 71.79 | 67.72 | 73.91 | 67.95 | 52.79 | 75.76 | 47.10 | 31.75 | 71.96 | 62.07 | 59.24 | 45.26 |
| | VUS-PR | **45.89** | 43.36 | 33.17 | 39.32 | 35.63 | 15.13 | 43.15 | 15.37 | 10.85 | 36.08 | 41.19 | 37.81 | 19.38 |
| MDT | Aff-F1 | 80.66 | 77.99 | 78.51 | 75.57 | **80.81** | 67.65 | 80.08 | 47.33 | 66.12 | 79.47 | 52.33 | 53.74 | 54.66 |
| | F1 | **59.36** | 53.70 | 48.39 | 51.70 | 49.14 | 27.71 | 54.88 | 19.54 | 25.46 | 49.40 | 43.84 | 38.10 | 20.75 |
| | Range-AUC-ROC | **86.59** | 63.94 | 58.98 | 58.78 | 59.00 | 42.44 | 77.01 | 43.65 | 41.41 | 75.61 | 54.86 | 53.22 | 41.90 |
| | Range-AUC-PR | **51.11** | 44.63 | 37.54 | 36.14 | 42.48 | 15.30 | 48.60 | 13.30 | 13.20 | 13.11 | 43.16 | 33.63 | 19.53 |
| | AUC-PR | 61.98 | 63.03 | 55.86 | 53.44 | 60.40 | 19.86 | **65.57** | 11.59 | 15.29 | 54.11 | 28.66 | 22.41 | 12.29 |
| | AUC-ROC | **90.74** | 79.04 | 75.65 | 73.19 | 74.79 | 60.30 | 86.67 | 53.82 | 56.44 | 84.55 | 60.26 | 63.92 | 54.51 |
| | VUS-ROC | **87.02** | 66.76 | 60.28 | 58.67 | 62.30 | 46.80 | 83.40 | 45.02 | 44.53 | 77.69 | 55.30 | 54.02 | 44.09 |
| | VUS-PR | **52.72** | 46.81 | 38.38 | 37.61 | 44.81 | 15.21 | 52.13 | 15.72 | 15.93 | 41.67 | 44.77 | 35.32 | 22.93 |
| KR | Aff-F1 | 85.44 | 84.22 | **89.55** | 82.24 | 79.56 | 71.80 | 85.47 | 61.94 | 70.99 | 79.52 | 64.78 | 69.38 | 58.11 |
| | F1 | **71.89** | 49.48 | 58.04 | 30.23 | 74.01 | 20.25 | 58.14 | 11.98 | 11.10 | 36.64 | 60.71 | 53.97 | 22.76 |
| | Range-AUC-ROC | **86.16** | 69.91 | 74.61 | 71.29 | 65.15 | 49.01 | 78.29 | 41.75 | 40.18 | 72.72 | 61.80 | 61.10 | 51.01 |
| | Range-AUC-PR | **59.64** | 46.95 | 51.59 | 40.75 | 37.53 | 13.25 | 52.83 | 6.04 | 7.44 | 35.22 | 51.69 | 43.07 | 18.99 |
| | AUC-PR | **72.91** | 63.55 | 66.72 | 55.39 | 56.78 | 28.19 | 67.47 | 8.10 | 7.01 | 53.60 | 41.09 | 32.21 | 10.18 |
| | AUC-ROC | **91.41** | 79.53 | 66.72 | 80.95 | 78.30 | 65.77 | 85.88 | 52.97 | 51.25 | 82.15 | 75.16 | 74.45 | 63.58 |
| | VUS-ROC | **86.77** | 70.82 | 75.99 | 73.93 | 67.81 | 47.06 | 79.00 | 43.04 | 41.97 | 74.65 | 58.77 | 60.70 | 47.51 |
| | VUS-PR | **58.58** | 45.90 | 51.41 | 43.32 | 38.23 | 19.10 | 51.60 | 8.49 | 7.94 | 36.18 | 54.17 | 43.31 | 24.19 |
| Energy | Aff-F1 | **71.37** | 64.38 | 60.20 | 63.84 | 66.37 | 65.85 | 66.00 | 47.07 | 43.39 | 66.85 | 55.85 | 62.03 | 57.65 |
| | F1 | **37.71** | 31.54 | 31.71 | 31.66 | 33.22 | 33.08 | 33.95 | 12.63 | 12.05 | 34.81 | 34.83 | 34.39 | 35.12 |
| | Range-AUC-ROC | **62.93** | 55.78 | 46.82 | 52.12 | 53.54 | 55.25 | 61.56 | 45.39 | 31.52 | 61.39 | 51.06 | 52.64 | 52.64 |
| | Range-AUC-PR | 33.24 | 33.47 | 28.81 | 30.70 | 31.10 | 30.59 | 38.17 | 21.77 | 19.24 | 35.25 | 42.14 | 45.19 | **43.89** |
| | AUC-PR | 39.82 | 37.81 | 38.05 | 27.51 | 33.75 | 32.85 | **42.05** | 17.69 | 14.02 | 34.25 | 21.55 | 21.17 | 21.69 |
| | AUC-ROC | **69.53** | 62.33 | 60.54 | 63.38 | 66.54 | 61.31 | 68.36 | 48.75 | 38.68 | 66.70 | 60.80 | 60.32 | 61.14 |
| | VUS-ROC | **61.46** | 54.37 | 46.03 | 51.15 | 53.10 | 53.04 | 59.47 | 45.93 | 31.56 | 58.31 | 51.50 | 53.61 | 53.07 |
| | VUS-PR | 35.15 | 34.18 | 29.46 | 31.04 | 31.68 | 30.35 | 38.61 | 22.57 | 19.69 | 34.41 | 42.57 | **46.03** | 44.30 |
| Weather | Aff-F1 | **80.84** | 69.01 | 75.46 | 76.17 | 72.56 | 73.68 | 80.58 | 42.80 | 49.22 | 77.17 | 47.70 | 54.06 | 64.91 |
| | F1 | 47.44 | 35.29 | 46.42 | 50.00 | 40.16 | 43.13 | **51.58** | 11.14 | 15.59 | 49.60 | 42.94 | 49.21 | 40.41 |
| | Range-AUC-ROC | 80.61 | 61.95 | 73.37 | 75.55 | 71.43 | 72.54 | **83.11** | 45.41 | 43.11 | 80.47 | 54.12 | 56.69 | 57.80 |
| | Range-AUC-PR | **50.88** | 29.86 | 43.20 | 44.31 | 41.37 | 43.40 | 50.58 | 18.06 | 18.85 | 49.81 | 46.37 | 49.65 | 47.47 |
| | AUC-PR | 49.16 | 29.80 | 48.87 | 49.91 | 44.12 | 49.71 | 47.56 | 17.08 | 16.71 | **53.39** | 31.16 | 35.44 | 25.02 |
| | AUC-ROC | 81.49 | 66.37 | 80.86 | 81.22 | 74.47 | 79.60 | 81.10 | 47.90 | 47.11 | **82.02** | 64.47 | 67.81 | 67.71 |
| | VUS-ROC | 80.40 | 61.03 | 73.22 | 75.08 | 70.03 | 71.71 | **81.91** | 45.56 | 43.32 | 79.97 | 54.16 | 56.45 | 57.38 |
| | VUS-PR | **50.76** | 30.00 | 43.21 | 44.35 | 41.30 | 43.47 | 50.09 | 18.33 | 19.17 | 50.13 | 46.58 | 49.66 | 47.13 |

