# OpenReview forum: "Towards a Multimodal Foundation Model for Time Series Analysis"
_ICLR.cc/2026/Conference — Submitted to ICLR 2026_

### Official Review · Reviewer_3z9H · 2025-10-18

**Soundness:** 3
**Presentation:** 3
**Contribution:** 3
**Rating:** 6
**Confidence:** 4

**Summary:**

This paper presents HORAI, a frequency-enhanced multimodal foundation model for time series analysis. Specifically, HORAI integrates two core components: a frequency-guided cross-modality encoder that leverages the correspondence between modality-specific information and different frequency components of time series to effectively fuse multiple modalities, and a time-frequency decoder that incorporates frequency information into a MoE router to improve pattern discrimination and generation. Experimental results show that HORAI achieves state-of-the-art zero-shot performance on time series forecasting and anomaly detection tasks, demonstrating strong task versatility and generalization.

**Strengths:**

1. The paper presents a creative and technically sound approach by introducing a frequency-guided cross-modality encoder. This design captures frequency-domain relationships across multiple modalities, allowing the model to align modality-specific representations through shared frequency patterns. Such integration not only enhances multimodal fusion but also enables the model to better capture both global and local temporal dependencies that are often overlooked in standard time-domain fusion methods.
2. The proposed combination of a frequency-guided encoder and a time-frequency decoder integrated with a Mixture-of-Experts (MoE) router demonstrates careful architectural planning. The encoder effectively bridges modality-specific representations, while the decoder utilizes frequency information to improve pattern discrimination and generation. This modular design makes HORAI versatile, interpretable, and easily extendable to a wide range of time series analysis tasks, including forecasting, classification, and anomaly detection.
3. The experimental section provides convincing evidence that HORAI achieves state-of-the-art performance on several benchmark datasets for time series forecasting and anomaly detection. Notably, its zero-shot capability highlights the generality of the proposed foundation model, showing robustness to unseen tasks and modalities. The consistent improvement across multiple datasets underscores both the effectiveness and broad applicability of the approach.

**Weaknesses:**

1. While the model demonstrates strong empirical performance, the paper lacks sufficient theoretical grounding or analytical explanations for why frequency-guided fusion enhances cross-modal representation learning. Without formal justification or interpretability studies, it remains unclear how much of the observed improvement stems from the proposed frequency-guided mechanism versus other architectural factors.
2. The experimental results, although promising, do not include detailed ablation studies to quantify the contribution of each key component—such as the frequency-guided encoder, MoE router, or specific frequency-selection strategies. Similarly, the paper does not provide sensitivity analysis regarding key hyperparameters, leaving open questions about the model’s robustness and reproducibility.
3. The computational complexity of HORAI, particularly its frequency-domain transformations and multimodal fusion layers, may pose scalability challenges for long time series or real-time applications. The paper does not clearly report training or inference costs, memory requirements, or optimization difficulties, which are crucial for understanding its feasibility in large-scale or resource-constrained environments.

**Questions:**

See weaknesses section

---

> ### Author Response · Authors · 2025-11-23
> **Response to Reviewer 3z9H (Part I)**
>
> Many thanks to Reviewer 3z9H for providing thorough and insightful comments. We have revised our paper accordingly.
>
> **W1:** Analysis of the frequency-guided fusion
>
> **Visualization Analysis**: We employ t-SNE to visualize the learned embeddings of the Energy dataset. As shown $\underline{\text{in Figure 8 of the revised paper}}$, the model learns to map modalities to their corresponding frequency time series components: Image features align closely with Mid-to-High Frequency time series components, while Text features cluster with Low-Frequency components. This visual evidence confirms the effectiveness of our frequency-guided alignment mechanism.
>
>
> **Frequency-guided mechanism**: To validate whether the performance gains stem specifically from our frequency-guided alignment strategy, we conduct two ablation variants: 1) removing the frequency-based alignment module entirely (w/o Freq-Align); and 2) swapping modalities by fusing low-frequency time series with images and by fusing mid-to-high frequency time series with text (modality exchange). The observed performance degradation in both variants demonstrates the effectiveness and necessity of our proposed frequency-guided alignment mechanism.
>
>
> |  | HORAI | W/O freq-align | modality exchange |
> | --- | --- | --- | --- |
> | Metrics | MSE / MAE | MSE / MAE | MSE / MAE |
> | Agriculture  | 0.236 / 0.332 | 0.266 / 0.347 | 0.292 / 0.352
> | Climate | 0.867 / 0.741 | 0.938 / 0.786 | 1.321 / 0.856
> | Energy | 0.250 / 0.358 | 0.290 / 0.398 | 0.292 / 0.402 |
> | Environment | 0.307 / 0.393 | 0.325 / 0.396 | 0.360 / 0.426
>
> **W2:** Ablation study of the key components and hyper-parameter analysis
>
> **Frequency-guided Encoder:** As suggested, we conduct ablation analysis on the contributions of individual modalities and the frequency-based fusion $\underline{\text{in the Appendix F.2 of the revised paper}}$, including: (a) text + time series, (b) image + time series, and (c) text + image + time series without frequency-based alignment (W/O freq-align).
>
> |  | (a) text + time series  | (b) image + time series | (c) W/O freq-align | HORAI |
> | --- | --- | --- | --- | --- |
> | Metrics | MSE / MAE | MSE / MAE | MSE / MAE | MSE / MAE |
> | Agriculture | 0.271 / 0.349 | 0.312 / 0.360 | 0.266 / 0.347 | 0.236 / 0.332
> | Climate | 1.102 / 0.828 | 0.982 / 0.797 | 0.928 / 0.786 | 0.867 / 0.741 |
> | Energy | 0.295 / 0.405 | 0.306 / 0.415 | 0.290 / 0.398 | 0.250 / 0.358 |
> | Environment | 0.344 / 0.412  | 0.320 / 0.395 | 0.325 / 0.396 | 0.307 / 0.393 |
>
>
> **MoE Router Ablation**： In the original paper, we have conducted the ablation study of the MoE router in $\underline{\text{the Section 4.4}}$, including removing frequency information from the router (W/O router), and replacing the Time-Frequency MoE-FFN with a standard FFN (W/O MoE-FFN).
>
>
>
> **Specific Frequency-selection Strategies**: As suggested, regarding the frequency selection strategy, we conduct the following experiments:
> (1) Threshold Analysis: We utilize a frequency threshold $\alpha$ to separate low-frequency components from mid-to-high frequency components. We have conducted a sensitivity analysis in $\underline{\text{Appendix F.1 of the revised paper}}$.
>
> (2) Modality Exchange: $\underline{\text{In Section 4.4 of the original paper}}$, we have conducted a "Modality Exchange" experiment. This involved aligning and fusing the low-frequency components of time series with images and the mid-to-high frequency components of time series with text, validating the effectiveness of our proposed alignment method.
>
> (3) Decomposition Method Replacement: As suggested, we replace the FFT-based frequency decomposition with a Moving Average approach to validate the effectiveness of the frequency domain strategy. The results are presented in the table below.
>
>
>
> |  | HORAI | Moving Average |
> | --- | --- | --- |
> | Metrics | MSE / MAE | MSE / MAE |
> | Agriculture  | 0.236 / 0.332 | 0.260 / 0.340 |
> | Climate | 0.867 / 0.741 | 0.984 / 0.785 |
> | Energy | 0.250 / 0.358 | 0.277 / 0.368 |
> | Environment | 0.307 / 0.393 | 0.327 / 0.398 |
>
>
>
> **Hyper-parameter analysis**: As suggested, we conduct hyper-parameter sensitivity analysis $\underline{\text{in the Appendix F.1 of the revised paper}}$，including frequency threshold $\alpha$ and the number of selected experts $K$.
>
>
> |  |  α﻿=0.01 | α﻿=0.05 | α﻿ = 0.25 | α﻿=0.5 |
> | --- | --- | --- | --- | --- |
> | Metrics | MSE | MSE | MSE | MSE |
> | Agriculture | 0.245 | **0.236** | 0.255 | 0.277 |
> | Climate | 0.868 | **0.867** | 1.054 | 1.200 |
> | Energy | 0.260 | **0.250** | 0.342 | 0.335 |
> | Environment | 0.313 | **0.307** | 0.332 | 0.333 |
>
> |  | K=1 | K=2 | K=3 | K=4 |
> | --- | --- | --- | --- | --- |
> | Metrics | MSE | MSE | MSE | MSE |
> | Agriculture | 0.258 | 0.236 | **0.232** | 0.252 |
> | Climate | 1.062 | **0.867** | 0.884 | 0.896 |
> | Energy | 0.262 | **0.250** | 0.260 | 0.265 |
> | Environment | 0.320 | **0.307** | 0.315 | 0.326 |

---

> ### Author Response · Authors · 2025-11-23
> **Response to Reviewer 3z9H (Part II)**
>
> **W3:** Computational complexity and efficiency analysis
>
>
>
> **Computational Complexity:**
>
> Given our adoption of a channel-independent strategy, we analyze the computational complexity of HORAI's key components based on a univariate time series of length  $L$:
> - Frequency-domain transform: We employ the Fast Fourier Transform (FFT) to convert the time series from the time domain to the frequency domain, which has a complexity of $O(L log L)$.
> - Multimodel fusion layers: For time series and image fusion, let $E_{ts} \in \mathbb{R}^{N_{ts} \times D_{model}}$ and $E_{image} \in \mathbb{R}^{N_{image} \times D_{model}}$ denote the representations of time series and images, respectively. We utilize Flow-attention [1] for alignment, resulting in a complexity of $O((N_{ts}+N_{image})\cdot D_{model})$. Similarly, for time series and text fusion, with the text representation $E_{text} \in \mathbb{R}^{N_{text} \times D_{model}}$, the alignment by Flow-attention yields a complexityis $O((N_{ts}+N_{text})\cdot D_{model})$.
>
> Overall, the complexity of these components remains low, ensuring both efficiency and scalability for multimodal modeling.
>
> [1] Flowformer: Linearizing Transformers with Conservation Flows
>
>
>
> **Efficiency Analysis:** In the original paper, we have provided a detailed efficiency analysis in $\underline{\text{the Appendix D}}$, covering parameters, MACs, and inference time. Compared to existing multimodal time-series-specific models and unimodal time series foundation models, HORAI demonstrates competitive efficiency. Regarding training cost, training HORAI on two NVIDIA A800 GPUs takes approximately 30 hours per epoch.

---

> > ### Comment · Reviewer_3z9H · 2025-11-27
> >
> > I think my concerns are well addressed by the author's additional experiments.

---

> ### Author Response · Authors · 2025-11-28
> **Sincere Gratitude from Authors**
>
> Dear Reviewer 3z9H,
>
> Many thanks for your prompt response and your positive support. We would like to once again express our sincere gratitude for taking the time to review our paper and for providing such insightful and invaluable comments.
>
> Best regards,
>
> The Authors

---

### Official Review · Reviewer_rxHD · 2025-10-27

**Soundness:** 3
**Presentation:** 3
**Contribution:** 2
**Rating:** 4
**Confidence:** 4

**Summary:**

This paper presents an initial step towards developing multimodal foundation models for time series analysis by addressing the challenges of data scarcity and multimodal integration. The core contributions include the construction of MM-TS, a large-scale multimodal pre-training dataset that integrates time series, text, and image modalities across six diverse domains, containing over one billion time points. Furthermore, the authors propose HORAI, a frequency-enhanced multimodal foundation model, which features a Frequency-guided Cross-Modality Encoder for effective fusion by aligning low-frequency components with text and mid-to-high-frequency components with vision, and a Time-Frequency Decoder that utilizes a Mixture-of-Experts (MoE) Feed-Forward Network (FFN) and a frequency-informed router for enhanced generalization. After pre-training on MM-TS, HORAI achieves state-of-the-art zero-shot performance on time series forecasting and anomaly detection tasks, demonstrating strong versatility and generalization ability.

**Strengths:**

1. The pursuit of a multimodal foundation model for time series analysis is a timely and valuable research direction, which addresses the limitations of existing unimodal time series foundation models. The construction of the large-scale, multimodal MM-TS dataset specifically tackles the critical challenge of data scarcity and provides a solid base for future research.

2. The proposed HORAI architecture is well-designed, particularly the Frequency-guided Cross-Modality Encoder that introduces a theoretically sound mechanism to fuse heterogeneous modalities by aligning them to different frequency components of the time series data (text with low-frequency, image with mid-to-high-frequency).

3. Relatively Complete Experiments: The experimental section is relatively comprehensive, evaluating the model on two key tasks, forecasting and anomaly detection, across numerous real-world datasets. The impressive state-of-the-art zero-shot performance compared to both unimodal foundation models and full-shot time-series-specific models demonstrates the strong generalization ability of HORAI.

**Weaknesses:**

1. The text and image modalities in MM-TS are derived from the numerical time series data (LLM-generated descriptions and line-plot visualizations ), which fundamentally means they do not introduce genuinely new external information. This limits the novelty and contribution of the multimodal data generation, essentially functioning as merely augmented representations of the time series itself. Given the reliance on LLMs for text generation, the authors must provide an evaluation of the fidelity and quality of the generated text descriptions.

2. The paper refers to using a "pre-trained text encoder" and a "pre-trained vision encoder", but the specific models are not identified. This lack of transparency makes reproduction difficult. Furthermore, a crucial ablation study comparing the impact of using different, representative encoders (e.g., different LLMs or Vision Transformers) is missing, which is necessary to confirm the robustness of the HORAI framework regardless of the underlying encoder choice.

3. The claim in Section 2.1 that existing multimodal methods "cannot generalize to new scenarios through zero-shot inference" is questionable, as foundational models like ChatTime are known to perform zero-shot forecasting, and a direct comparison is necessary. More critically, the forecasting results in Table 1, where Zero-Shot Time Series Models frequently outperform Full-shot Time Series Models, is highly counter-intuitive. This abnormal finding suggests that the small sample sizes of the evaluation datasets might be preventing the Full-shot models from converging properly. A comprehensive analysis and experimental investigation of this phenomenon must be provided.

4. Given the high complexity of anomaly evaluation, assessing performance solely based on the three threshold-independent metrics (AUC-ROC, VUS-ROC, and VUS-PR) is insufficient for a complete analysis. To align with the standard practice in the anomaly detection field, the authors should provide a multi-dimensional comparison utilizing a broader and more diverse set of evaluation metrics.

**Questions:**

Please refer to Weaknesses.

---

> ### Author Response · Authors · 2025-11-23
> **Response to Reviewer rxHD (Part I)**
>
> We would like to sincerely thank Reviewer rxHD for providing a detailed review and insightful suggestions. We have revised our paper accordingly.
>
> **W1:** The novelty and contribution of the multimodal data
>
> We wish to emphasize that our contribution extends beyond the construction of a dataset; it fundamentally introduces a generalizable multimodal pre-training paradigm.
> - **Paradigm Shift**:  Multimodal time series field is currently impeded by the scarcity of large-scale aligned data, restricting existing methods to either small-scale multimodal or large-scale unimodal training. To break this limitation, we propose a novel paradigm utilizing an endogenous construction strategy that uniquely integrates three distinct modalities (time series, images, text). Its core innovations are as follows:
>   * **Tri-modal Synergy**: Unlike previous works restricted to bi-modal alignment (e.g., Time-Text or Time-Image), our paradigm unifies Semantic information from Text and Visual Perception from Images. By synthesizing these complementary modalities, we transform unimodal numerical signals into comprehensive multimodal knowledge, enhancing time series analysis from a multi-view perspective.
>   * **Endogenous Construction**: This strategy enables the generation of massive-scale, well-aligned multimodal datasets. Compared to loose exogenous information, endogenous generation ensures significantly stronger cross-modal correlations and intrinsic alignment, making it uniquely suited for learning generalized representations during pre-training.
>
> - **Dataset Construction**: Building on this paradigm, we construct MM-TS, the first large-scale multimodal dataset for time series analysis, containing 1 billion time points across six domains and covering time series, text, and images.
>   * **Methodological Value**: By designing sophisticated prompts for LLMs and rendering time series as line plots, we enrich raw time series with rich semantic and visual contexts. MM-TS provides the essential multi-source supervision signals required for effective cross-modal alignment.
>
>
> We also revise $\underline{\text{the introduction of the revised paper}}$ to better highlight the contributions.
>
>
> **The Fidelity and Quality of the Text Descriptions:**
> - **Structured Prompting**: As illustrated in Figure 6 of the paper, we design a highly structured prompt that explicitly constrains the LLM's output, ensuring the content is relevant to the input time series.
> - **Quality Evaluation**：We randomly sampled 5% of the pretraining data and used GPT-4 to evaluate the quality of the generated text across three dimensions (scored from 1 to 5): 1) Whether the text accurately describes time-series patterns such as periodicity and trends; 2) Whether the provided external factors are logically consistent with the domain context; 3) Whether the text contains concrete, specific details rather than generic content. The average scores on these three dimensions are 4.72, 4.81, and 4.65, respectively, demonstrating the reliability and high quality of the generated text.
>
>
> **W2:** The ablation analysis of different text encoders and vision encoders
>
> For the text encoder, we adopt Qwen 0.5B, and for the visual encoder, we use ViT-Base.
>
> To evaluate the model’s performance with different encoders, we conduct additional experiments by replacing both text and visual encoders $\underline{\text{in the Appendix F.3 of the revised paper}}$. Considering time and computational constraints, we select encoders with relatively small parameter sizes. Specifically, the text encoders include **GPT2-large, LLaMA3-1B, and Qwen2.5-1.5B**, while the visual encoder comparison uses **Swin Transformer**.
>
>
> As shown in the table, for a given text encoder, models with larger parameter sizes tend to perform slightly better, and employing more advanced architectures (e.g., Qwen, and LLaMA) generally yields further improvements. In the comparison of visual encoders, ViT and Swin Transformer achieve similar overall forecasting performance.
>
>
>
>
>
> |  | HORAI | GPT2-large | Llama3-1B | Qwen-1.5B | Swin Transformer-Base |
> | --- | --- | --- | --- | --- | --- |
> | Metrics | MSE  / MAE | MSE / MAE | MSE / MAE | MSE / MAE | MSE / MAE |
> | Agriculture | 0.236 / 0.332 | 0.258 / 0.352 | 0.228 / 0.315 | 0.237 / 0.334 | 0.230 / 0.325 |
> | Climate | 0.867 / 0.741 | 0.913 / 0.842 | 0.874 / 0.752 | 0.850/ 0.732 | 0.876 / 0.761 |
> | Energy | 0.250 / 0.358 | 0.265 / 0.372 | 0.245 / 0.350 | 0.229 / 0.342 | 0.254 / 0.364 |
> | Environment | 0.307 / 0.393 | 0.325/ 0.398 | 0.310 / 0.398 | 0.304 / 0.392 | 0.300 / 0.388 |

---

> ### Author Response · Authors · 2025-11-23
> **Response to Reviewer rxHD (Part II)**
>
> **W3:** Discussion with ChatTime and Explanation of Why the Zero-Shot Time Series Model Outperforms the Full-Shot Time Series Model
>
>
> We appreciate the reviewer for pointing out the inaccuracy regarding the zero-shot capabilities of existing multimodal methods (e.g., ChatTime). We have rectified this statement in $\underline{\text{Section 2.1 of the revised paper}}$. The revised text now reads:
>
> "Although these methods achieve competitive performance, most require retraining and extensive parameter tuning for each dataset, lacking zero-shot inference capabilities. While ChatTime enables direct zero-shot inference, it suffers from precision loss due to data discretization and lacks rich multimodal characterizations."
>
>
> We further provide a clarified discussion comparing ChatTime and HORAI, and we include ChatTime as a baseline in our multimodal experiments.
>
> **Discussion with ChatTime：**
> - **Model perspective**: HoRAI is **specifically architected as a multimodal foundation model** integrating time series, images, and text. It leverages modality-specific encoders to extract distinct features and employs a novel frequency-enhanced alignment to explicitly fuse these representations from multiple perspectives. In contrast,  ChatTime **adapts general-purpose LLMs** for time series analysis. While leveraging LLMs' inherent reasoning abilities for time series analysis offers generalization, discretizing continuous numerical values into textual tokens leads to precision loss, making it difficult to capture time series patterns.
>
> - **Data perspective**: HORAI is pretrained on a large-scale multimodal dataset incorporating aligned text and images. These modalities capture diverse characterizations of temporal dynamics from multiple perspectives and simultaneously introduce some external context, providing relevant supervision that improves generalization. However, ChatTime relies only on simple prompts such as “Please predict the following sequence,” which offer limited text regarding the specific time series characteristics.
>
> **Experiment Results:** As shown in the table below, HORAI outperforms ChatTime on most datasets, demonstrating the stronger generalization ability of our multimodal pretraining paradigm.
>
> |  | HORAI | ChatTime |
> | --- | --- | --- |
> | Metrics | MSE / MAE | MSE / MAE |
> | Agriculture | 0.236 / 0.332 | 0.369 / 0.410 |
> | Climate | 0.867 / 0.741 | 1.860 / 1.106 |
> | Energy | 0.250 / 0.358 | 0.247 / 0.352 |
> | Environment | 0.307 / 0.393 | 0.359 / 0.456 |
> | Social Good | 0.792 / 0.451 | 1.069 / 0.503 |
> | Traffic | 0.176 / 0.293 | 0.596 / 0.610 |
> | EWJ | 0.591 / 0.542 | 0.887 / 0.641 |
> | KR | 0.551 / 0.448 | 0.565 / 0.455 |
> | MDT | 0.373 / 0.434 | 0.496 / 0.479 |
>
>
>
>
>
> **Phenomenon Explanation**: We would like to clarify that it is a reasonable and expected phenomenon for a time series foundation model in a zero-shot setting to outperform time-series-specific models in a full-shot setting. The reasons are twofold: 1）**Strong Generalization via Pre-training**: HORAI benefits significantly from its large-scale learnable parameters and extensive pre-training on massive multi-source and multi-modal datasets. This enables the model to capture diverse and complex time series patterns, resulting in robust generalization capabilities.  2）**Precedents in Foundation Models**: This phenomenon is widely observed in the landscape of foundation models. For instance, MOIRAI, a unimodal time series foundation model, demonstrated that its zero-shot capabilities could surpass the performance of state-of-the-art full-shot time-series-specific models at the time.
>
>
> **Convergence Analysis of Full-shot Models**: We conduct a detailed analysis using two representative models: TATS and GPT4MTS on the small-scale Traffic dataset. We visualize both the training and validation losses (as shown in $\underline{\text{Figure 7 of revised paper}}$). The training and validation loss curves show a steady decrease followed by stabilization. This indicates that the time-series-specific models converged properly.
>
> **W4:** More metrics of anomaly evaluation
>
> Thank you for your suggestion. We add additional evaluation metrics for anomaly detection, including Standard F1, Affiliation F1 (Aff-F1), AUC-PR, Range-AUC-ROC, and Range-AUC-PR. The results for all metrics are reported in $\underline{\text{Table 12 of the revised paper}}$. Overall, the experiments show that HORAI achieves state-of-the-art anomaly detection performance across most metrics, demonstrating its strong generalization ability on these datasets.

---

> ### Author Response · Authors · 2025-11-28
> **Looking forward to your feedback**
>
> Dear Reviewer rxHD,
>
> We would like to express our sincere gratitude for your time and efforts in reviewing our paper, which has inspired further improvements to our paper.
>
> **As the discussion period is approaching its end**, we were wondering whether our response addressed all of your concerns satisfactorily.
> If you have any further concerns or questions, please do not hesitate to inform us, and we will be more than happy to address them promptly.
>
>
> All the best,
>
> Authors

---

### Official Review · Reviewer_oNEh · 2025-10-30

**Soundness:** 2
**Presentation:** 3
**Contribution:** 2
**Rating:** 4
**Confidence:** 4

**Summary:**

This work proposes MM-TS, a large-scale multimodal time-series dataset that covers multiple domains with extensive time points. Based on MM-TS, the paper introduces HORAI, a multimodal time-series foundation model trained on MM-TS, featuring adaptive fusion selection and frequency MoE decoding. The evaluation results demonstrate the effectiveness of HORAI on multimodal time-series forecasting across multiple domains.

**Strengths:**

1. This work presents MM-TS, a large-scale multimodal time-series pre-training dataset that could potentially benefit future research on multimodal time-series forecasting.

2. The proposed HORAI is a multimodal time-series foundation model trained on MM-TS, incorporating multimodal fusion (mainly frequency-enhanced). The evaluations show strong zero-shot forecasting performance across domains beyond those included in its training data.

**Weaknesses:**

1. While constructing MM-TS involves substantial effort, the dataset itself does not contain newly collected time-series data. Most of the data in MM-TS are from well-known unimodal datasets such as Monash or PEMS. MM-TS mainly aggregates these datasets and performs data augmentation for additional modalities, which may reduce the novelty and overall contribution of the dataset.

2. In constructing MM-TS, the prompts specify the start and end times for generating textual summaries. Does this imply that models trained on MM-TS must rely on a fixed historical lookback window, since the summaries correspond to fixed time ranges? Clarifying this would help understand whether MM-TS supports flexible temporal contexts.

3. Regarding the modality alignment in the frequency domain, it is unclear why the vision component is specifically aligned with mid-to-high frequency parts. Could this introduce noise instead of meaningful signals? How is $\alpha$ properly set to avoid this issue in a foundation model (or is one $\alpha$ good for all cases)? Moreover, how does this frequency-based alignment differ from a simpler time-series decomposition approach, which is easier to interpret and does not rely on complex-valued representations?

4. For the forecasting evaluation on Time-MMD, this dataset already includes textual information, but its text structure may differ from that in MM-TS. Does the evaluation directly use the original Time-MMD text, or is it reconstructed following the MM-TS text generation process?

5. Although the paper includes results with and without additional modalities, more detailed ablation studies would strengthen the analysis of multimodal benefits—especially regarding the frequency alignment. For example, comparing forecasting with (a) text only, (b) image only, and (c) text + image without frequency-based alignment could provide clearer insights into how each modality contributes to performance improvements.

**Questions:**

Please refer to the weakness.

---

> ### Author Response · Authors · 2025-11-23
> **Response to Reviewer oNEh (Part I)**
>
> Many thanks to Reviewer oNEh for providing valuable comments. We have revised our paper accordingly.
>
> **W1:** The novelty and contribution of the dataset
>
> We appreciate the reviewer’s recognition of our effort. However, we wish to emphasize that our contribution extends beyond the construction of a dataset; it fundamentally introduces a generalizable multimodal pre-training paradigm.
> - **Paradigm Shift**:  Multimodal time series field is currently impeded by the scarcity of large-scale aligned data, restricting existing methods to either small-scale multimodal or large-scale unimodal training. To break this limitation, we propose a novel paradigm utilizing an endogenous construction strategy that uniquely integrates three distinct modalities (time series, images, text). Its core innovations are as follows:
>   * **Tri-modal Synergy**: Unlike previous works restricted to bi-modal alignment (e.g., Time-Text or Time-Image), our paradigm unifies Semantic information from Text and Visual Perception from Images. By synthesizing these complementary modalities, we transform unimodal numerical signals into comprehensive multimodal knowledge, enhancing time series analysis from a multi-view perspective.
>   * **Endogenous Construction**: This strategy enables the generation of massive-scale, well-aligned multimodal datasets. Compared to loose exogenous information, endogenous generation ensures significantly stronger cross-modal correlations and intrinsic alignment, making it uniquely suited for learning generalized representations during pre-training.
>
> - **Dataset Construction**: Building on this paradigm, we construct MM-TS, the first large-scale multimodal dataset for time series analysis, containing 1 billion time points across six domains and covering time series, text, and images.
>   * **Methodological Value**: By designing sophisticated prompts for LLMs and rendering time series as line plots, we enrich raw time series with rich semantic and visual contexts. MM-TS provides the essential multi-source supervision signals required for effective cross-modal alignment.
>
> We also revise $\underline{\text{the introduction of the revised paper}}$ to better highlight the contributions.
>
> **W2:** The flexible temporal contexts of MM-TS
>
> We design MM-TS to support flexible temporal contexts. Instead of generating text for every specific sliding window (which would create redundancy), we generate interval-wise text descriptions covering broad intervals (e.g., 1000 steps). During training, we sample input time series sub-sequences (e.g., 576 steps) that fall within these larger intervals. This design effectively decouples the text generation window from the specific input lookback window, ensuring that the model is not reliant on a fixed historical window. It learns to utilize the broader semantic context provided by the text to guide the representation of any sub-sequence within that range.

---

> ### Author Response · Authors · 2025-11-23
> **Response to Reviewer oNEh (Part II)**
>
> **W3:** The vision component align with the mid-to-high frequency parts, $\alpha$ value selection，and other time series decomposition approach.
>
> **Frequency-based Alignment:** We align the visual modality with the mid-to-high frequency components, leveraging the inherent inductive bias of vision representation models, which are well-suited for capturing local structural patterns. These patterns intrinsically correspond to short-term fluctuations and variations reflected in these mid-to-high frequency components.
>
> To validate this design, we conduct ablation experiments: 1) removing frequency-based alignment and fusing images with the entire time series (W/O freq-align), and 2) swapping modalities by fusing low-frequency time series with images and by fusing mid-to-high frequency time series with text (modality exchange). The results show performance drops in both variants, demonstrating the effectiveness of aligning images with mid-to-high frequency time series.
>
>
> |  | HORAI | W/O freq-align | modality exchange |
> | --- | --- | --- | --- |
> | Metrics | MSE / MAE | MSE / MAE | MSE / MAE |
> | Agriculture  | 0.236 / 0.332 | 0.266 / 0.347 | 0.292 / 0.352
> | Climate | 0.867 / 0.741 | 0.938 / 0.786 | 1.321 / 0.856
> | Energy | 0.250 / 0.358 | 0.290 / 0.398 | 0.292 / 0.402 |
> | Environment | 0.307 / 0.393 | 0.325 / 0.396 | 0.360 / 0.426
>
>
>
> **Noise Issues and Alpha Value Selection:**
> We would like to clarify that the mid-to-high frequency components in the time series represent meaningful signals, not noise. Since we set the $\alpha$  to 0.05 for the FFT-based frequency-domain decomposition, these retained components capture structured local patterns and periodic variations rather than random noise. This alpha value is consistently maintained throughout both pre-training and downstream evaluation.
>
> To assess the impact of $\alpha$ on prediction performance, we conduct a sensitivity analysis $\underline{\text{in the Appendix F.1 of the revised paper}}$. The results in the table below show that setting alpha to 0.05 yields good prediction results. A larger value pushes most information into high-frequency components, amplifying noise-like patterns. Setting it too small introduces low-frequency information, affecting image and time series alignment.
>
> |  |  α﻿=0.01 | α﻿=0.05 | α﻿ = 0.25 | α﻿=0.5 |
> | --- | --- | --- | --- | --- |
> | Metrics | MSE | MSE | MSE | MSE |
> | Agriculture | 0.245 | 0.236 | 0.255 | 0.277 |
> | Climate | 0.868 | 0.867 | 1.054 | 1.200 |
> | Energy | 0.260 | 0.250 | 0.342 | 0.335 |
> | Environment | 0.313 | 0.307 | 0.332 | 0.333 |
>
> **Simple Time Series Decomposition:**
>
> Our choice of FFT-based decomposition instead of simpler methods (e.g., moving average) is motivated by improving the robustness to outliers. Moving average techniques are susceptible to outliers, where single anomaly may distort the trend, leading to inaccurate decomposition. FFT avoids this by explicitly separating signal components based on global frequency characteristics, ensuring that transient anomalies (high-frequency) do not corrupt the extraction of the long-term trend (low-frequency).
>
> We also replace FFT-based decomposition with moving average decomposition, and the prediction performance declined, confirming the advantage of the FFT-based decomposition.
>
> |  | HORAI | Moving Average |
> | --- | --- | --- |
> | Metrics | MSE / MAE | MSE / MAE |
> | Agriculture  | 0.236 / 0.332 | 0.260 / 0.340 |
> | Climate | 0.867 / 0.741 | 0.984 / 0.785 |
> | Energy | 0.250 / 0.358 | 0.277 / 0.368 |
> | Environment | 0.307 / 0.393 | 0.327 / 0.398 |

---

> ### Author Response · Authors · 2025-11-23
> **Response to Reviewer oNEh (Part III)**
>
> **W4:** The use of TimeMMD in the evaluation
>
>
> We directly use the original Time-MMD text for evaluation. Although its text structure differs from that of MM-TS, the two share important similarities:
> - Both describe the underlying time series dynamics, thus conveying related semantic information;
> - The MM-TS text is generated using LLMs, which inherently encode broad world knowledge and introduce external contextual cues—for example, the prompt in Figure 6 includes instructions such as “Consider the broader contextual factors affecting this dataset, including seasonal variations…”, which partially overlap with the type of information present in Time-MMD.
>
> Thanks to these semantic similarities and HORAI’s strong generalization capability, the model achieves good performance on Time-MMD in the zero-shot setting.  To further assess HORAI’s adaptability, we perform a 10% few-shot evaluation on Time-MMD. The results show consistent improvements, demonstrating that HORAI not only generalizes well but can also rapidly adapt to new textual formats with minimal supervision.
>
> |  | HORAI (zero-shot) | HORAI (10% few-shot) |
> | --- | --- | --- |
> | Metrics | MSE / MAE | MSE / MAE |
> | Agriculture | 0.236 / 0.332 | 0.210 / 0.289 |
> | Climate | 0.867 / 0.741 | 0.857 / 0.739 |
> | Energy | 0.250 / 0.358 | 0.234 / 0.325 |
> | Environment | 0.307 / 0.393 | 0.274 / 0.379 |
>
>
> **W5:** The ablation analysis of the specific modalities
>
>
> We thank the reviewer for the suggestion. We add an ablation analysis on specific modality $\underline{\text{in the Appendix F.2 of the revised paper}}$, including: (a) text + time series, (b) image + time series, and (c) text + image + time series without frequency-based alignment (W/O freq-align).
>
> The results in the table show that both the image and text modalities contribute to performance improvements, with varying impacts across datasets. For datasets with clear trends, such as Agriculture and Energy, the text modality contributes more, whereas for datasets with local fluctuations, such as Climate, the image modality contributes more.
>
>
>
> |  | (a) text + time series  | (b) image + time series | (c) W/O freq-align | HORAI |
> | --- | --- | --- | --- | --- |
> | Metrics | MSE / MAE | MSE / MAE | MSE / MAE | MSE / MAE |
> | Agriculture | 0.271 / 0.349 | 0.312 / 0.360 | 0.266 / 0.347 | 0.236 / 0.332
> | Climate | 1.102 / 0.828 | 0.982 / 0.797 | 0.928 / 0.786 | 0.867 / 0.741 |
> | Energy | 0.295 / 0.405 | 0.306 / 0.415 | 0.290 / 0.398 | 0.250 / 0.358 |
> | Environment | 0.344 / 0.412  | 0.320 / 0.395 | 0.325 / 0.396 | 0.307 / 0.393 |

---

> ### Author Response · Authors · 2025-11-28
> **Looking forward to your feedback**
>
> Dear Reviewer oNEh,
>
> We would like to express our sincere gratitude for your time and efforts in reviewing our paper, which has inspired further improvements to our paper.
>
> **As the discussion period is approaching its end**, we were wondering whether our response addressed all of your concerns satisfactorily.
> If you have any further concerns or questions, please do not hesitate to inform us, and we will be more than happy to address them promptly.
>
>
> All the best,
>
> Authors

---

### Official Review · Reviewer_8MWe · 2025-11-01

**Soundness:** 3
**Presentation:** 2
**Contribution:** 3
**Rating:** 4
**Confidence:** 3

**Summary:**

The paper proposes a time series analysis framework using foundation models under multi-modal learning scenario. It builds a large-scale datasets and proposes a corresponeding method for model learning.

**Strengths:**

1. Time series analysis is always a valuable field for research, then the topic of this paper is solid.
2. This paper proposes a large scale dataset, which should be regarded as a contribution for the community.
3. Overall, the proposed framework is reasonable, and empirical performance shows its effectiveness.

**Weaknesses:**

1. I am confused about the main goal of this draft. Is the dataset contribution the main part? or the proposed method. If it is the former one, probably a dataset tract fits more and the current version needs more dataset based analysis. If the later one, the proposed method is relatively straightforward, while the multi-modal scenario is a good point which requires great experimental effort.
2. The draft looks not ready yet, especially some formats.  For example, tab1 and 2 are not well adapted with the main text. Figs are not informative with small font size.
3. Time series analysis is a huge research area, adding more ablation analysis will strongly support this draft, while current empirical focus is more on performance comparison aspect.

**Questions:**

Please check above section.

---

> ### Author Response · Authors · 2025-11-23
> **Response to Reviewer 8MWe (Part I)**
>
> We would like to sincerely thank Reviewer 8MWe for providing detailed and insightful comments. We have revised our paper accordingly.
>
> **W1:** Main Goal of the paper
>
> We appreciate the reviewer’s insightful comment regarding the positioning of our paper. We would like to clarify that this work represents a **comprehensive multimodal pre-training solution**, where the dataset and the method are mutually dependent and reinforcing. As the field is in a nascent stage, a standalone dataset or method is insufficient. Therefore, our main goal is to propose a unified **Paradigm, Dataset, and Method**.
>
> To address your specific concerns, we provide clarifications on the Main Contributions of our paper below:
>
> *   **Multimodal Pre-training Paradigm & Dataset:** Existing methods are mostly limited to small-scale multimodal training (e.g., GPT4MTS, TimeMMD) or to large-scale unimodal pre-training (e.g., Timer, MOIRAI), which restricts their generalization ability. The lack of large-scale, well-aligned datasets keeps multimodal pre-training for time series in a nascent stage. To break this limitation, we propose a novel paradigm that utilizes three distinct modalities (time series, images, text) by an endogenous construction strategy with LLMs. This approach synthesizes large-scale aligned multimodal data to enhance time series analysis from a **multi-view perspective**, leveraging endogenous pre-training to adapt to exogenous modalities, thereby enabling good **zero-shot generalization in downstream scenarios**. Based on this paradigm, we construct MM-TS, the first large-scale multimodal time series dataset, covering six domains with up to 1 billion time points.
>
> *   **Foundation Model:** Based on the paradigm and dataset, we design the multimodal foundation model HORAI that utilizes frequency-view fusion to align low and mid-to-high frequency components of time series with text and image respectively, effectively leveraging the respective strengths of these endogenous modalities to enhance time series analysis. Additionally, we introduce a frequency-informed MoE decoder where frequency patterns explicitly guide expert routing to enhance generalization.
>
> Through this unified paradigm, dataset, and method, we take an early step toward the development of this emerging field.
>
> We revise the $\underline{\text{Introduction in the revised paper}}$ to clearly articulate the main goal.
>
> **Novelty of HORAI (Why it is not Straightforward)**
> *   **Frequency-View Multimodal Fusion:** Unlike standard fusion methods, HORAI innovatively decomposes time series into frequency bands. It aligns low-frequency components with images and mid-high-frequency components with text. This structure-aware fusion more effectively leverages each modality and enhances time series analysis.
> *   **Frequency-Informed MoE Routing:** We introduce a novel mechanism where frequency patterns explicitly guide the MoE routing. This ensures that expert networks specialize in different temporal dynamics, significantly boosting the model's generalization capability during pre-training.

---

> ### Author Response · Authors · 2025-11-23
> **Response to Reviewer 8MWe (Part II)**
>
> **W2:** Some formatting issues
>
> We thank the reviewer for pointing out these formatting issues. We have revised the model type names in Tables 1 and 2 to ensure better consistency with the main text. Additionally, we have improved Figure 1 by enlarging the font size of the MM-TS text samples for better clarity, standardizing the font styles, and correcting the typo 'Time-Freqency' to 'Time-Frequency'.
>
> **W3:** More ablation analysis
>
> Thank you for your suggestions. We add more ablation analysis in the revised paper, specifically:
> (1) An ablation study on the contribution of individual modalities and alignment strategies, provided in $\underline{\text{in the Appendix F.2 of the revised paper}}$.
>
> | | HORAI | text + time series  | image + time series | W/O freq-align | modality exchange |
> | --- | --- | --- | --- | --- | --- |
> | Metrics | MSE / MAE | MSE / MAE | MSE / MAE | MSE / MAE | MSE / MAE |
> | Agriculture | 0.236 / 0.332 | 0.271 / 0.349 | 0.312 / 0.360 | 0.266 / 0.347 | 0.292 / 0.352
> | Climate | 0.867 / 0.741 | 1.102 / 0.828 | 0.982 / 0.797 | 0.928 / 0.786 | 1.321 / 0.856 |
> | Energy | 0.250 / 0.358 | 0.295 / 0.405 | 0.306 / 0.415 | 0.290 / 0.398 | 0.292 / 0.402 |
> | Environment | 0.307 / 0.393 | 0.344 / 0.412  | 0.320 / 0.395 | 0.325 / 0.396 | 0.360 / 0.426 |
>
> (2) An ablation analysis of the text encoder and vision encoder $\underline{\text{in the Appendix F.3 of the revised paper}}$.
>
>
> |  | HORAI | GPT2-large | Llama3-1B | Qwen-1.5B | Swin Transformer-Base |
> | --- | --- | --- | --- | --- | --- |
> | Metrics | MSE  / MAE | MSE / MAE | MSE / MAE | MSE / MAE | MSE / MAE |
> | Agriculture | 0.236 / 0.332 | 0.258 / 0.352 | 0.228 / 0.315 | 0.237 / 0.334 | 0.230 / 0.325 |
> | Climate | 0.867 / 0.741 | 0.913 / 0.842 | 0.874 / 0.752 | 0.850/ 0.732 | 0.876 / 0.761 |
> | Energy | 0.250 / 0.358 | 0.265 / 0.372 | 0.245 / 0.350 | 0.229 / 0.342 | 0.254 / 0.364 |
> | Environment | 0.307 / 0.393 | 0.325/ 0.398 | 0.310 / 0.398 | 0.304 / 0.392 | 0.300 / 0.388 |
>
> (3) A sensitivity analysis for key hyper-parameters $\underline{\text{in the Appendix F.1 of the revised paper}}$, such as the frequency threshold $\alpha$, and the number of selected experts $K$.
>
> |  |  α﻿=0.01 | α﻿=0.05 | α﻿ = 0.25 | α﻿=0.5 |
> | --- | --- | --- | --- | --- |
> | Metrics | MSE | MSE | MSE | MSE |
> | Agriculture | 0.245 | **0.236** | 0.255 | 0.277 |
> | Climate | 0.868 | **0.867** | 1.054 | 1.200 |
> | Energy | 0.260 | **0.250** | 0.342 | 0.335 |
> | Environment | 0.313 | **0.307** | 0.332 | 0.333 |
>
> |  | K=1 | K=2 | K=3 | K=4 |
> | --- | --- | --- | --- | --- |
> | Metrics | MSE | MSE | MSE | MSE |
> | Agriculture | 0.258 | 0.236 | **0.232** | 0.252 |
> | Climate | 1.062 | **0.867** | 0.884 | 0.896 |
> | Energy | 0.262 | **0.250** | 0.260 | 0.265 |
> | Environment | 0.320 | **0.307** | 0.315 | 0.326 |

---

> ### Author Response · Authors · 2025-11-28
> **Looking forward to your feedback**
>
> Dear Reviewer 8MWe,
>
> We would like to express our sincere gratitude for your time and efforts in reviewing our paper, which has inspired further improvements to our paper.
>
> **As the discussion period is approaching its end**, we were wondering whether our response addressed all of your concerns satisfactorily.
> If you have any further concerns or questions, please do not hesitate to inform us, and we will be more than happy to address them promptly.
>
>
> All the best,
>
> Authors

---

### Author Response · Authors · 2025-12-03
**Summary of the Rebuttal**

Dear AC, SAC, PC,

We are sorry to hear about the recent OpenReview bug issue, and we fully support the proposed remedy actions.

We sincerely thank the new AC, the original AC, the PC, SAC, and all reviewers for your time and effort in coordinating and evaluating our submission. The feedback from each reviewer has been invaluable throughout the entire process. To facilitate the decision-making process,  we believe it is necessary to briefly summarize the status of our paper.

1. **Reviewers' Responses:** During the rebuttal, Reviewer 3z9H explicitly stated that their concerns were well-addressed. Reviewers 8MWe, oNEh, and rxHD did not provide further responses.

***

2. **Main Content of the Paper:** Multimodal time series field is currently impeded by **the scarcity of large-scale aligned data**, restricting existing methods to either small-scale multimodal or large-scale unimodal training. To break this bottleneck, we pioneer a novel **multimodal pretraining paradigm** that leverages time series together with their derived image and text. This paradigm enhances time series analysis from a multi-view perspective, enabling good zero-shot generalization in downstream scenarios. Based on this paradigm, we construct MM-TS, **the first billion-scale multimodal time series dataset** across six domains, and propose **HORAI, a frequency-enhanced multimodal foundation model** with a Frequency-guided Cross-Modality Encoder and a Time-Frequency Decoder.

***

3. **Reviewers' Opinions:** We are pleased to see that the reviewers agree our paper **creative and well-designed** (Reviewer rxHD, 8MWe, 3z9H), **the topic is solid and valuable** (Reviewer 8MWe, rxHD), **benefits future research** (Reviewer 8MWe, oNEh, rxHD), **complete experiments and SOTA performance** (Reviewer rxHD,3z9H).



4. **Reviewers' Concerns:** In response to the reviewers’ valuable and constructive concerns, we have carefully revised our paper and conducted additional experiments accordingly. Here is the summary of the main concerns and our responses.

| Concerns | Revisions | Reviewer |
|--------| ------ | ------ |
| **Clarification of main goal** | We clarify that the main goal is to propose a **unified Paradigm, Dataset, and Method**. To support this, we elaborate on the specific contribution of each component and revise the introduction of the paper. | 8MWe |
| **Ablation study** | We add ablation studies: 1）remove specific modalities and replace alignment strategies (Appendix F.2),  2）replace different text encoder and vision encoder (Appendix F.3)， 3）sensitivity analysis about the frequency threshold $\alpha$ and the number of selected experts $K$ (Appendix F.1). | 8MWe, oNEh, rxHD, 3z9H |
| **Dataset Contribution** |  We clarify that the contribution **extends beyond the dataset to a generalizable tri-modal pre-training paradigm**. Based on this paradigm, we construct MM-TS, **the first large-scale multimodal dataset** for time series analysis.  | oNEh, rxHD|
| **Discussion with ChatTime** | We discuss the differences between HORAI and ChatTime from different perspectives (Appendix E), and add ChatTime as a baseline in Table 1. | rxHD|
| **Generalization to Time-MMD Textual Formats** | We clarify the shared underlying semantics between MM-TS and TimeMMD，and add few-shot experiments, validating model robustness and adaptability to diverse textual formats. | oNEh |
| **Additional anomaly evaluation metrics** | We add more evaluation metrics for anomaly detection in Table 12, showing SOTA results across most metrics. | rxHD|
| **Efficiency analysis** |  We analyze the computational complexity of key module and provide efficiency analysis in Appendix D. | 3z9H |



We once again sincerely appreciate your time and the additional effort required to evaluate our paper. We hope that this summary assists the new AC in reducing their workload when making final decision.

Best regards,

Authors

---

### Meta-Review · Area_Chair_pLMY · 2026-01-08

**Summary:**

Overall, this paper presents two primary contributions: the construction of a multimodal dataset and the development of a multimodal foundation model tailored for that dataset. However, the reviewers found the breakthrough potential of both components to be limited. Specifically: 1) the dataset itself does not contain newly collected time-series data; and 2) the performance gains derived from modal alignment remain unclear. Consequently, I recommend rejection.

**Reviewer Concerns:**

Reviewer 8MWe expressed confusion regarding the paper's primary objective: is the core contribution the dataset or the proposed methodology? Furthermore, the reviewer noted that neither aspect was sufficiently substantiated.

1. Dataset Contributions

Reviewers oNEh and rxHD observed that while constructing the MM-TS dataset clearly involved significant effort, the dataset itself lacks original, newly collected time-series observations, relying instead on existing data.

2. Methodological Innovation

Reviewers rxHD and 3z9H raised concerns regarding the significance of modal alignment within the frequency domain. Specifically, it remains unclear why the visual components should be aligned with the mid-to-high frequency segments of the time-series data.

**Reviewer Scores:**

Although the authors provided responses to these points during the rebuttal phase, the issues raised require further clarification and more substantial improvements than what was provided.

---

### Decision · Program_Chairs · 2026-01-26

Reject